EMBO
Molecular Medicine

# Inhibition of PIKfyve prevents myocardial apoptosis and hypertrophy through activation of SIRT3 in obese mice

Helene Tronchere[1,2,†], Mathieu Cinato[1,2,†], Andrei Timotin[1,2], Laurie Guitou[1,2], Camille Villedieu[3], Helene Thibault[3], Delphine Baetz[3], Bernard Payrastre[1,2], Philippe Valet[1,2], Angelo Parini[1,2], Oksana Kunduzova[1,2] (iD) & Frederic Boal[1,2,*] (iD)

## Abstract

PIKfyve is an evolutionarily conserved lipid kinase that regulates pleiotropic cellular functions. Here, we identify PIKfyve as a key regulator of cardiometabolic status and mitochondrial integrity in chronic diet-induced obesity. *In vitro*, we show that PIKfyve is critical for the control of mitochondrial fragmentation and hypertrophic and apoptotic responses to stress. We also provide evidence that inactivation of PIKfyve by the selective inhibitor STA suppresses excessive mitochondrial ROS production and apoptosis through a SIRT3-dependent pathway in cardiomyoblasts. In addition, we report that chronic STA treatment improves cardiometabolic profile in a mouse model of cardiomyopathy linked to obesity. We provide evidence that PIKfyve inhibition reverses obesity-induced cardiac mitochondrial damage and apoptosis by activating SIRT3. Furthermore, treatment of obese mice with STA improves left ventricular function and attenuates cardiac hypertrophy. In contrast, STA is not able to reduce isoproterenol-induced cardiac hypertrophy in SIRT3.KO mice. Altogether, these results unravel a novel role for PIKfyve in obesity-associated cardiomyopathy and provide a promising therapeutic strategy to combat cardiometabolic complications in obesity.

**Keywords** apoptosis; cardiac hypertrophy; mitochondria; PIKfyve; SIRT3
**Subject Categories** Cardiovascular System; Metabolism

## Introduction

Global increase in rates of obesity-associated cardiovascular complications poses a major challenge to overall population health (Battiprolu *et al*, 2012). Obesity has been linked to a spectrum of changes in metabolic status and cardiac phenotype with reduced contractility, left ventricular hypertrophy and heart failure (Battiprolu *et al*, 2012). Given the limited capacity of the heart for regeneration, the progression of these abnormalities in obese patients poses a major threat. However, the specific mechanisms through which obesity triggers the decline of cardiac function remain unclear. A fragile balance between survival and death exists in cardiac cells undergoing pathologic stress, and activation of apoptotic machinery leads to premature cell death and heart failure progression (Barouch *et al*, 2006). This loss of cardiomyocytes may be secondary to mitochondrial dysfunction caused by chronic exposure to reactive oxygen species (ROS) in obese conditions (Bournat & Brown, 2010; Tsutsui *et al*, 2011). However, to date, effective therapeutic tools that control both metabolic and cardiac abnormalities in obese patients are elusive.

The evolutionarily conserved lipid kinase PIKfyve that synthesizes PI5P and $PI(3,5)P_2$ has been implicated in a diverse range of cellular processes, including cell proliferation, migration, tyrosine kinase receptor signaling, and membrane trafficking (Shisheva, 2008). PIKfyve is ubiquitously expressed in mammals, including in cardiac tissue (Ikonomov *et al*, 2013), and the total knockout is embryonic lethal in mice (Ikonomov *et al*, 2011). It contains a FYVE domain which binds to PI3P on endosomes and is responsible for its intracellular localization (Sbrissa *et al*, 2002b). Expression of a PIKfyve dominant negative mutant (Ikonomov *et al*, 2001), epigenetic or pharmacological inhibition of PIKfyve (Jefferies *et al*, 2008) induces the formation of enlarged endosomal vacuoles, indicating its critical role in the maintenance of the endo-lysosomal membrane homeostasis. Recently, a new potent and highly selective PIKfyve inhibitor has been characterized, shedding light on a new role for PIKfyve in inflammation and autoimmune diseases (Cai *et al*, 2013). Indeed, this inhibitor, known as apilimod or STA-5326 (referred to STA throughout this study), shows great potency to reduce the production of pro-inflammatory cytokines and TLR signaling in dendritic cells and monocytes. STA has been tested in patients with

1   INSERM U1048 I2MC, Toulouse, Cedex 4, France
2   Université Paul Sabatier, Toulouse, France
3   CarMeN Laboratory, Inserm U1060, Univ-Lyon, Université Claude Bernard Lyon 1, Bron, France
    *Corresponding author. Tel: +33 531224117; E-mail: frederic.boal@inserm.fr
    †These authors contributed equally to this work

   

rheumatoid arthritis (Krausz *et al*, 2012), psoriasis (Wada *et al*, 2012), and Crohn's disease (Billich, 2007). To date, there are no reports concerning the effects of STA on cardiac and metabolic disorders.

In this study, we unravel a critical role of PIKfyve in the regulation of cardiometabolic status in obesity-induced phenotype. We provide evidence that chronic inhibition of PIKfyve by STA attenuates obesity-related cardiometabolic phenotype by reducing mitochondrial oxidative stress and apoptosis through the deacetylase SIRT3. Therefore, these data pave the way to new promising therapeutic strategies to prevent cardiometabolic complications in obesity.

# Results

## STA treatment attenuates hypertrophic response and mitochondrial ROS production in cardiomyoblasts

Oxidative and metabolic stresses are key factors in the pathogenesis of obesity-related diseases (Bournat & Brown, 2010; Tsutsui *et al*, 2011). To investigate whether PIKfyve activity is affected in conditions of metabolic or oxidative stress, we measured PI5P levels in H9C2 cells subjected to hypoxia-induced oxidative stress or 2-deoxy-D-glucose (2DG)-induced metabolic stress. We found a ~fivefold increase in PI5P levels in H9C2 cells in response to hypoxic (Fig 1A) or 2DG stimulations (Fig 1B) as measured by mass assay. This PI5P synthesis was totally abrogated by the selective PIKfyve inhibitor STA, providing the first evidence that stress-induced cellular responses are linked to PIKfyve activity. Interestingly, we found that basal levels of PI5P are refractory to STA treatment in cardiomyoblasts.

Cardiac hypertrophy is a potent predictor of cardiovascular risk in obesity (Battiprolu *et al*, 2012). To examine the potential role of the lipid kinase PIKfyve in hypertrophic responses to stress, we evaluated the effects of its pharmacological inhibition by STA on hypoxia-induced hypertrophy in cardiomyoblasts. As shown in Fig 1C, STA treatment of H9C2 cells induced the formation of enlarged vacuoles, a hallmark of PIKfyve inhibition (Ikonomov *et al*, 2001; Jefferies *et al*, 2008). As described by Dupuis-Coronas *et al* (2011) and others (Jefferies *et al*, 2008) in various cell types, the formation of these endosomal vacuoles did not impede H9C2 cell viability (Fig EV1A), even at high concentration. Strikingly, STA treatment abrogated hypoxia-induced hypertrophic responses as

shown by measuring the cell surface (Fig 1C) and quantification of the hypertrophic marker β-MHC (Fig 1D). Cell hypertrophy is closely linked to ROS production by mitochondria, a major site for ROS production (Sawyer *et al*, 2002); therefore, we next investigated whether PIKfyve inhibition affects hypoxia-induced mitochondrial ROS production. Remarkably, H9C2 cells treated with STA presented a reduced level of mitochondrial $O_2^-$ (Fig 1E, MitoSOX) and $H_2O_2$ (Fig 1E, MitoPY1) in response to hypoxic stress. Moreover, PIKfyve inhibition by STA attenuated 2DG-induced mitochondrial ROS production in H9C2 cells (Fig 1F). In order to confirm STA specificity toward PIKfyve, we resorted to siRNA-mediated silencing of PIKfyve in H9C2 cells. Silencing efficiency was monitored by qRT-PCR (Fig EV1B). Notably, we found that PIKfyve silencing prevented hypoxia-induced ROS production to the same extent as STA treatment (Fig EV1C). Interestingly, we demonstrated that in cells depleted for PIKfyve, STA has no further effect on hypoxia-induced ROS generation, validating PIKfyve as the target for the anti-oxidant properties of STA.

## PIKfyve inhibition prevents stress-induced cell apoptosis and mitochondrial structural damage

Mitochondrial damage and excessive ROS production may result in activation of apoptotic cascades and cell death (Tsutsui *et al*, 2011). As shown in Fig 2, in response to hypoxia, STA treatment of H9C2 cells attenuated apoptosis as shown by TUNEL staining (Fig 2A) and cleavage of caspase 3, a bona fide marker of apoptotic cascade activation (Fig 2B). Importantly, STA-dependent anti-apoptotic activity was confirmed in conditions of metabolic stress induced by 2DG (Fig 2C and D).

One of the hallmarks of apoptosis is the fragmentation of the mitochondrial network (Youle & Karbowski, 2005). Therefore, we next examined whether inhibition of PIKfyve could affect stress-induced mitochondrial fragmentation. While control cells harbored a typical elongated and interconnected mitochondrial network, both hypoxia (Fig 2E) and 2DG (Fig 2F) resulted in mitochondrial fragmentation. Strikingly, PIKfyve inhibition by STA prevented mitochondrial fragmentation induced by both oxidative (Fig 2E) and metabolic stresses (Fig 2F) suggesting STA-dependent preservation of mitochondrial integrity. Notably, depletion of PIKfyve by siRNA recapitulated STA effect on the preservation of mitochondrial structures upon hypoxic stress (Fig EV1D).

Mitochondrial fragmentation is mediated by recruitment of the small cytosolic GTPase dynamin-related protein 1 (Drp1) at the

---

**Figure 1.  Inhibition of PIKfyve by STA reduces cardiomyoblast hypertrophic response and mitochondrial ROS production.**

A    Rat H9C2 cardiomyoblasts were subjected to hypoxia (H) or kept in normoxia (N) in the presence of STA or vehicle only (DMSO). PI5P levels were measured by mass assay to address PIKfyve activation (left panel). Quantification of PI5P from independent experiments is shown on the right panels (*n* = 3–5).

B    Same as in (A) but cells were exposed to 2DG-induced metabolic stress.

C    H9C2 cells were subjected to hypoxia (H) to induce cell hypertrophy or kept in normoxia (N) in the presence of STA or vehicle only (DMSO). Phase contrast images are shown (left panel). Scale bar is 25 μm. Cell surface was quantified from 239 to 279 cells across three independent experiments.

D    qRT-PCR quantification of the expression level of the hypertrophic marker β-MHC from three to six independent experiments.

E, F    H9C2 cells were exposed to oxidative- (E) or 2DG-induced metabolic (F) stress as indicated and mitochondrial $O_2^-$ production or mitochondrial $H_2O_2$ were assessed using the MitoSOX Red fluorescent probe and MitoPY1 probe, respectively (left panels). Scale bar is 10 μm. Quantifications are shown on the right panels (*n* = 16–81).

Data information: Data are presented as mean ± SEM. Two-way ANOVA followed by Bonferroni's *post hoc* test: \*\*\*P < 0.001, \*\*P < 0.01, and \*P < 0.05 between indicated conditions. The exact *P*-values are specified in Appendix Table S1.

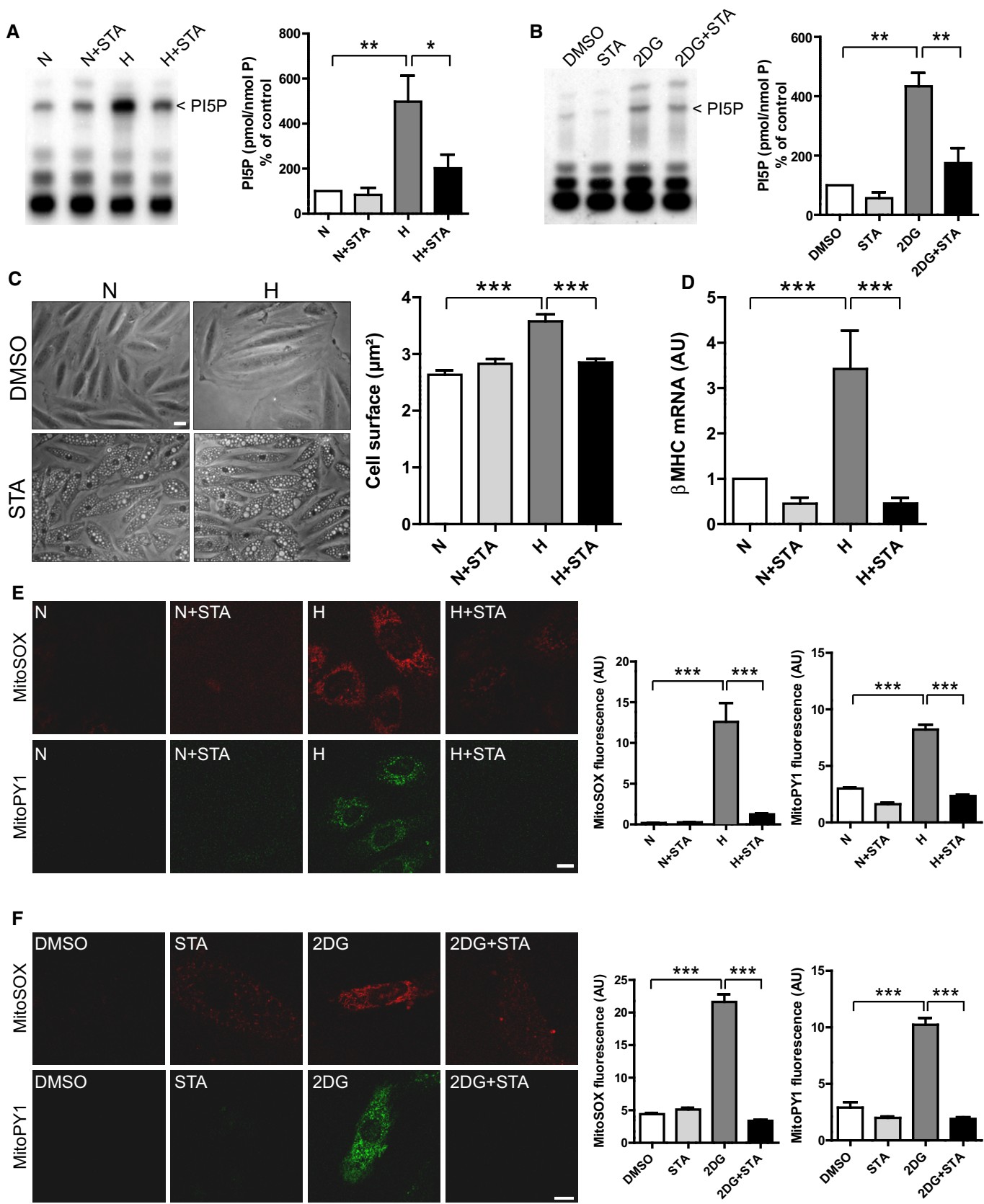

**Figure 1.**

**Figure 2.**

**Figure 2.  STA treatment prevents cardiomyoblast apoptotic cell death and preserves mitochondrial structures.**

A TUNEL staining of apoptotic H9C2 cells treated as indicated (left panel). Scale bar is 50 μm. Right panel shows quantification of apoptotic cells from three to four independent experiments.

B Cell lysates from H9C2 cells treated as in (A) were probed with the indicated antibodies (left panel). Quantification of caspase 3 cleavage is shown on the right panel from three to four independent experiments.

C TUNEL staining of apoptotic cells treated with 2-deoxy-ᴅ-glucose (2DG) in the presence or not of STA. Quantification is shown on the right panel from three to four independent experiments. Scale bar is 50 μm.

D Western blot of H9C2 cells treated as in (C). Results are from four independent experiments.

E, F H9C2 cells were treated as indicated, and live-stained with MitoTracker Red to assessed mitochondrial structures (left panels). Scale bar is 10 μm. Mitochondrial fragmentation was quantified on thresholded images using a dedicated ImageJ plugin. Quantification of mitochondrial fragmentation is shown on the right panels ($n$ = 10–51).

G, H Self-assembly of Drp1 at mitochondrial fission sites was monitored by immunofluorescence on treated cells (left panels). Scale bar is 10 μm. Quantification of the number of Drp1 punctae per cell is shown on the right panels ($n$ = 33–114).

Data information: Data are presented as mean ± SEM. Two-way ANOVA followed by Bonferroni's *post hoc* test: ***$P < 0.001$, **$P < 0.01$, and *$P < 0.05$ between indicated conditions.

Source data are available online for this figure.

active fission site on the surface of mitochondria, which can be followed by immunofluorescence (Frank *et al*, 2001; Smirnova *et al*, 2001). As shown in Fig 2G and H, both oxidative (Fig 2G) and metabolic (Fig 2H) stresses induced the self-assembly of Drp1 in H9C2 cells. In these conditions, STA treatment reduced Drp1 assembly, to the same extent as the Drp1-specific inhibitor Mdivi-1 (Fig 2G and H). siRNA-mediated depletion of PIKfyve also led to the reduction in Drp1 self-assembly induced by hypoxia (Fig EV1E), demonstrating the implication of PIKfyve in the control of mitochondrial dynamics.

### PIKfyve induces mitochondrial ROS production and apoptosis through a SIRT3-dependent pathway

In cardiac myocytes, the high density of mitochondria reflects the high energy demand needed to maintain contractile functions. Therefore, in order to maintain the redox cellular status and optimize the bioenergetic efficiency of the heart, the functioning of mitochondria is in turn tightly regulated. The NAD⁺-dependent lysine deacetylase SIRT3 has recently emerged has a key regulator of mitochondrial functions, through the control of the oxidative and metabolic status, mitochondrial dynamics, and apoptosis (Huang *et al*, 2010; McDonnell *et al*, 2015). SIRT3 is a nuclear-encoded protein and therefore needs to be translocated into the mitochondrial matrix to deacetylate its targets (Schwer *et al*, 2002). In order to test whether PIKfyve affects cardiac SIRT3, we first localized endogenous SIRT3 in H9C2 cells subjected to oxidative stress in the presence of STA. In control cells, SIRT3 was found mainly cytosolic (Fig 3A). Interestingly, a strong translocation of SIRT3 to the mitochondria was induced by STA treatment independently of stress stimuli (Fig 3A and B). This mitochondrial enrichment was confirmed biochemically by isolating mitochondria from control or STA-treated cells (Fig 3C and D). Importantly, the localization of the nuclear SIRT1 was not altered by STA treatment (Fig 3E), suggesting a specificity toward SIRT3. In order to confirm the results obtained with the pharmacological inhibition of PIKfyve, we examined SIRT3 localization in H9C2 cells silenced for PIKfyve expression using siRNA. As shown in Fig 3F, knockdown of PIKfyve in H9C2 cells resulted in a strong translocation of SIRT3 to the mitochondrion without changes in SIRT1 localization.

Next, in order to determine whether SIRT3 is involved in PIKfyve regulation of mitochondrial ROS generation and apoptosis, we resorted to its silencing using specific siRNA. Knockdown efficiency was confirmed by qRT-PCR (Fig EV2). We hypothesized that SIRT3 depletion may result in the loss of STA-induced anti-oxidant and anti-apoptotic activities. As shown in Fig 4, in conditions of metabolic stress, SIRT3 silencing totally prevented STA effects on mitochondrial ROS production (Fig 4A and B) and cell apoptosis (Fig 4C and D). These results point to an unprecedented role of PIKfyve in the control of mitochondrial ROS production, cell hypertrophy, and apoptosis through the control of SIRT3 pathway.

### STA treatment reduces cardiac hypertrophy and improves cardiac function in a mouse model of diet-induced obesity

Considering the *in vitro* effects of STA on cellular responses to oxidative and metabolic stresses, we next examined whether PIKfyve inhibition could improve cardiometabolic phenotype in a mouse model of chronic high fat diet (HFD)-induced obesity. As shown in Table 1, the exposure of mice to HFD for 12 months resulted in the development of glucose intolerance, insulin resistance, and morphometric changes as compared with normal diet (ND) fed mice. Echocardiographic analysis revealed ventricular dysfunction characterized by the decreased ejection fraction (EF) and fractional shortening (FS) and cardiac hypertrophy as shown by elevated LVPWd and IVSTd in HFD-fed mice as compared to ND-fed mice (Fig 5A–E). Analysis of heart weight-to-body weight ratio (HW/BW, Fig 5F), cardiac myocyte cross-sectional area (Fig 5G),

**Table 1.  Metabolic parameters of mice under ND or HFD feeding.**

| Parameters | ND | HFD |
|---|---|---|
| Body weight (g) | 42.8 ± 2.1 | 51.7 ± 1.8** |
| Fat mass (%) | 9.6 ± 0.8 | 19.6 ± 2.2*** |
| Glucose (mmol/l) | 10.2 ± 0.4 | 12.8 ± 0.8** |
| IGTT AUCglucose | 38,310 ± 842 | 45,722 ± 4,161* |

Body weight, fat mass, plasma glucose, area under the curve of intraperitoneal glucose tolerance test (IGTT AUCglucose) were evaluated in male C57BL/6J mice after 12 months HFD or ND feeding. $n$ = 6–14 per group. Data are means ± SEM; Student's *t*-test, *$P < 0.05$, **$P < 0.01$, and ***$P < 0.001$ versus ND-fed group.

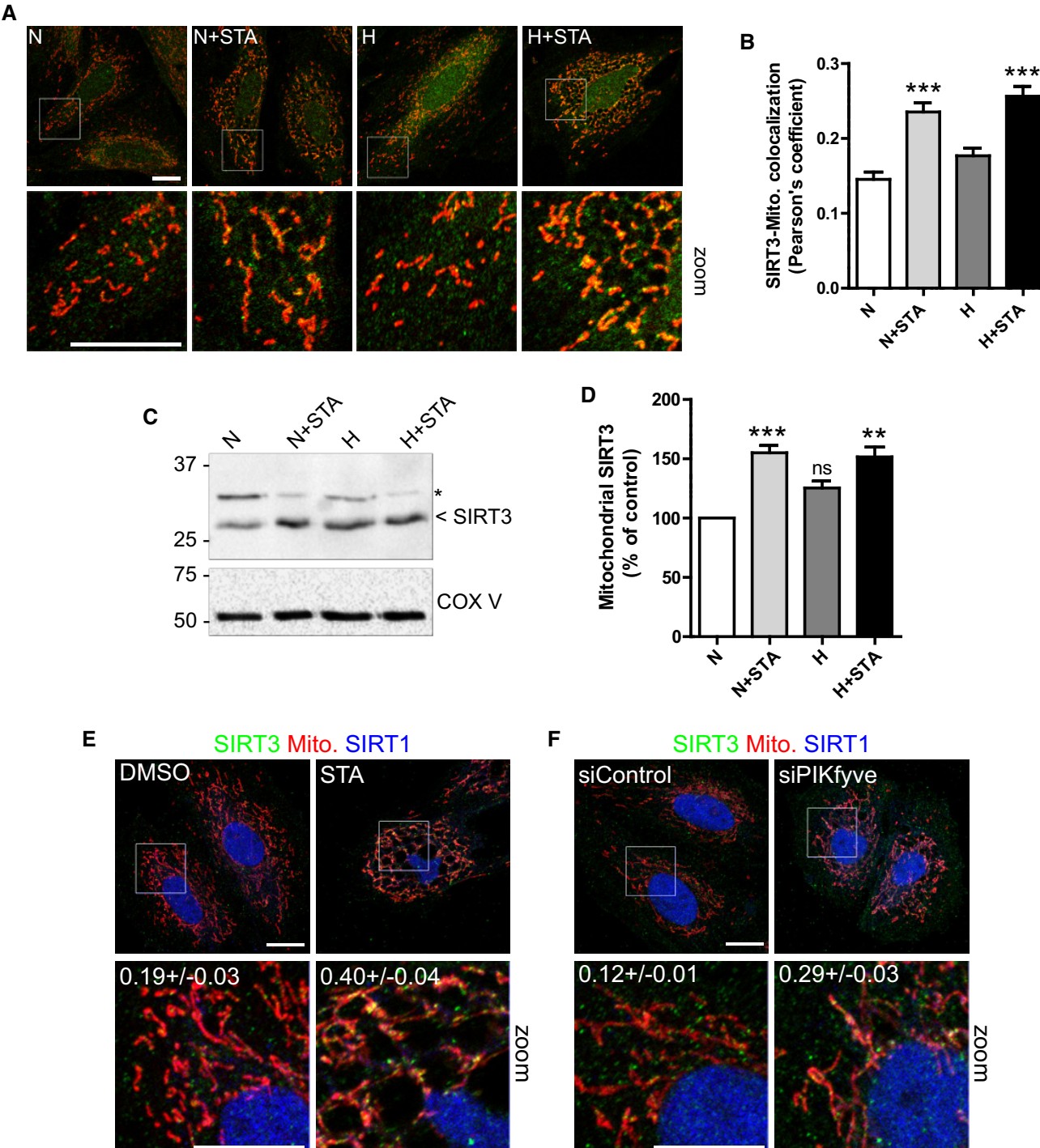

**Figure 3.  PIKfyve inhibition induces SIRT3 translocation to the mitochondria.**

A  H9C2 cells were treated as indicated, mitochondria were live-stained with MitoTracker Red (in red), and the cells were fixed and stained for endogenous SIRT3 (in green) and imaged by confocal microscopy. Scale bar is 10 μm.

B  Quantification of the colocalization between SIRT3 and mitochondria in treated cells from (A). Data are presented as mean ± SEM. Two-way ANOVA followed by Bonferroni's *post hoc* test: \*\*\*$P < 0.001$ as compared with control ($n = 69$–91).

C  Mitochondria were isolated from H9C2 cells treated as indicated and blotted for SIRT3 and COX V as a control. Asterisk indicates non-specific band.

D  Quantification of SIRT3 enrichment in mitochondria from (C). Data are presented as mean ± SEM. Two-way ANOVA followed by Bonferroni's *post hoc* test: \*\*$P < 0.01$ and \*\*\*$P < 0.001$ as compared with control ($n = 3$). ns, non-significant.

E  H9C2 cells were treated with STA as indicated, fixed and stained for endogenous SIRT3 (in green), SIRT1 (in blue). Mitochondria were live-stained with MitoTracker Red (in red). Scale bar is 10 μm. Pearson's coefficients are indicated in the zoomed boxes as mean ± SEM from three to four independent experiments.

F  H9C2 cells were transfected with a control siRNA (siControl) or with a siRNA targeting PIKfyve (siPIKfyve), fixed and stained as in (E). Scale bar is 10 μm.

Source data are available online for this figure.

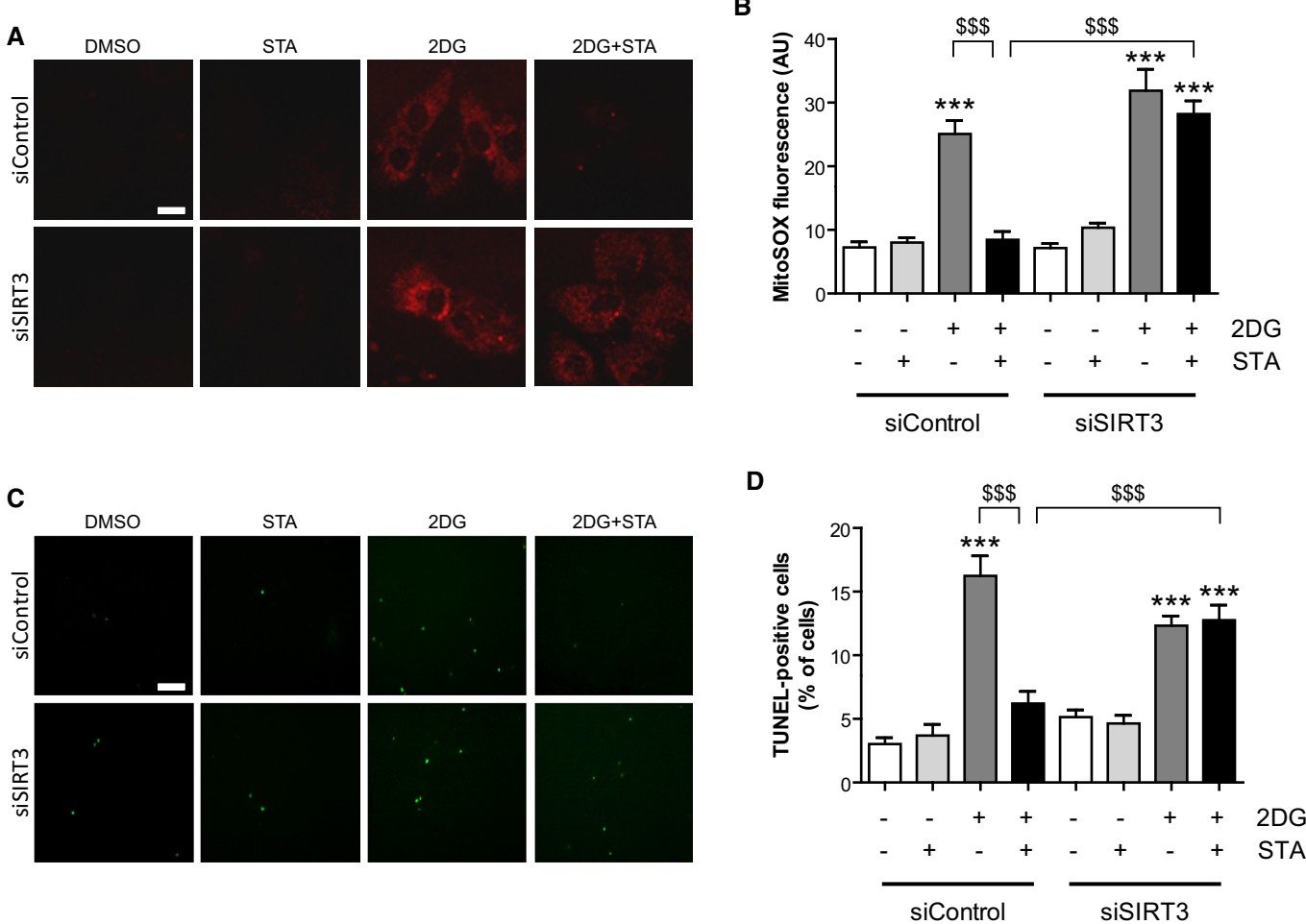

**Figure 4. Endogenous SIRT3 is required for STA anti-oxidant and anti-apoptotic properties.**

A  H9C2 cells were transfected with a control siRNA (siControl) or with a siRNA targeting SIRT3 (siSIRT3), and cells were treated as indicated. Mitochondrial $O_2^-$ production was assessed using the MitoSOX Red fluorescent probe. Scale bar is 10 μm.

B  Quantification from (A) (*n* = 75–157).

C  TUNEL staining of apoptotic cells treated as in (A). Scale bar is 50 μm.

D  Quantification of apoptotic cells from (C) (*n* = 4–7).

Data information: Data are presented as mean ± SEM. One-way ANOVA followed by Bonferroni's *post hoc* test: ***$P < 0.001$ as compared with control cells; $^{$$$}P < 0.001$ between indicated conditions.

and myocardial expression of hypertrophic markers β-MHC (Fig 5H) and BNP (Fig 5I) confirmed the induction of cardiac hypertrophy in HFD-fed mice as compared to ND-fed mice. In HFD-fed mice, PIKfyve inhibition prevented obesity-induced impairments in cardiac function and structure. Indeed, in HFD-fed mice, chronic treatment with STA improved cardiac function as shown by the increased EF and FS (Fig 5A–C). Compared to vehicle-treated HFD-fed mice, STA treatment reduced cardiac hypertrophy as shown by the decrease in the LVPWd and IVSTd (Fig 5D and E, respectively), HW/BW ratio (Fig 5F), cardiomyocyte cross-sectional area (Fig 5G), and myocardial expression of β-MHC and BNP (Fig 5H and I). Importantly, STA-dependent preservation of cardiac function was accompanied by a reduction in myocardial fibrosis as compared to vehicle-treated HFD-fed mice (Fig 5J). Moreover, STA treatment,

leading to reduced myocardial PI5P level (Fig EV3A), improved glucose tolerance as compared to vehicle-treated mice (Fig EV3B and C) without significant changes in body weight (44.8 g ± 3.3 in control versus 51.7 g ± 2.8 in STA-treated mice). In the same line, we found that STA did not change the amount of perigonadal adipose tissue (2.27% ± 0.16 in vehicle-treated mice versus 2.20% ± 0.11 in STA-treated mice, expressed as % of body weight), suggesting that STA treatment had no major effect on fat depot in obese mice. In addition, anti-glycemic activity of STA was associated with reduced levels of plasma triglycerides and myocardial content of lipid peroxide (LPO), an oxidative stress marker (Fig EV3D and E).

STA has been initially described for its anti-inflammatory properties (Cai *et al*, 2013). In cardiac tissue from HFD-fed mice, STA

treatment did not significantly affect myocardial expression level of key inflammatory factors including IL-1β, IL-12, IL-23, IL-6, TNF-α, and MCP1 (Fig EV4A). Moreover, no changes were detected in plasma IL-6 and TNF-α in STA-treated mice (Fig EV4B). This suggests that the cardioprotective effects of STA in cardiomyopathy linked to obesity are not due to its systemic anti-inflammatory properties.

### PIKfyve inhibition prevents obesity-induced oxidative stress, cardiac apoptosis, and mitochondrial damage

Based on our *in vitro* data, we next asked whether PIKfyve inhibition was able to affect obesity-induced oxidative stress and cardiac apoptosis. Chronic consumption of HFD resulted in enhanced production of mitochondrial ROS (Fig 6A and B) and

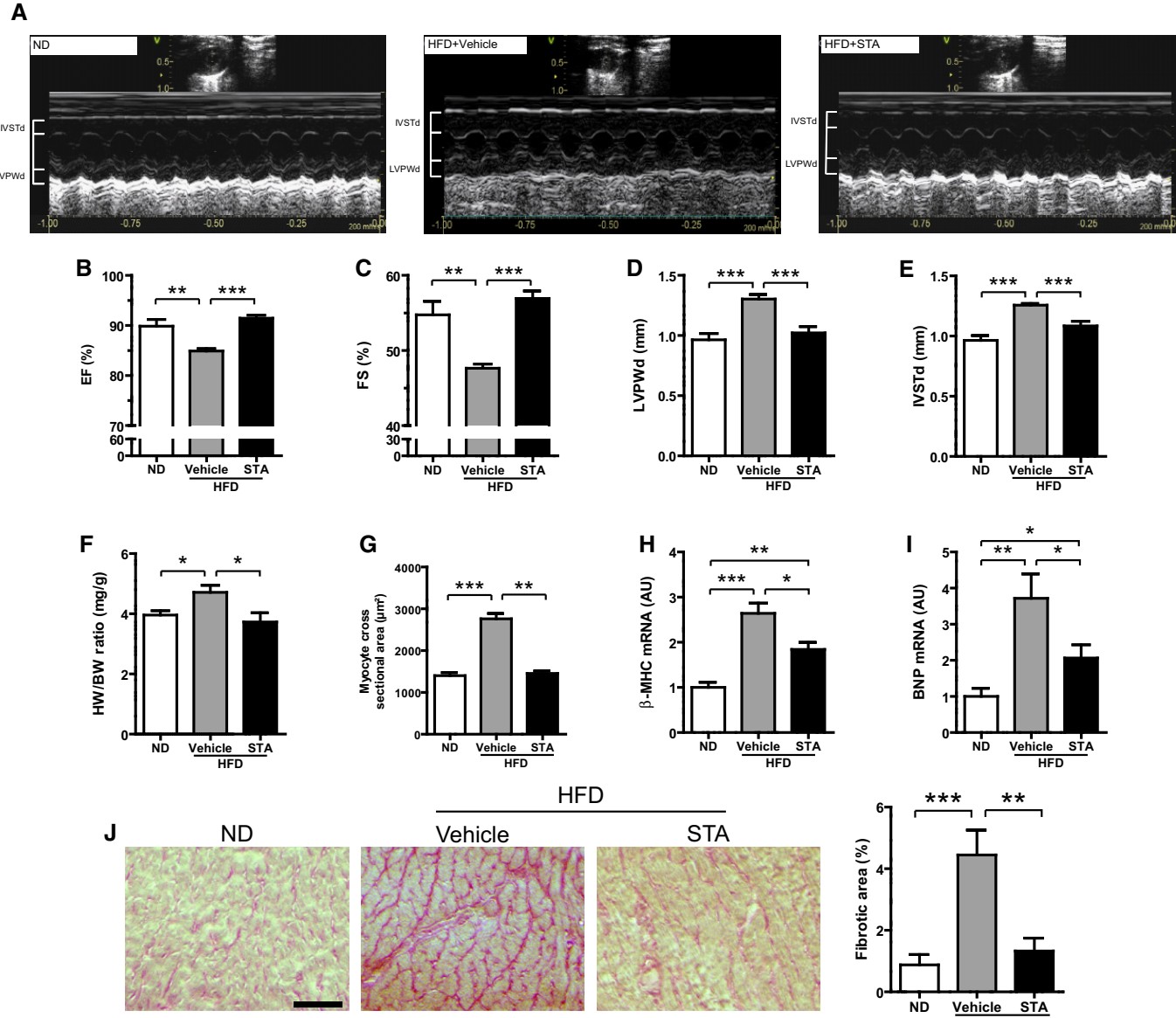

**Figure 5. PIKfyve inhibition reduces cardiac hypertrophy and improves cardiac function *in vivo*.**

A Representative 2D-M-Mode echocardiographic images of non-obese (ND) or obese (HFD) mice treated intraperitoneally with STA or vehicle only (Vehicle).

B–E Echocardiographic measures of ejection fraction (EF, B), fractional shortening (FS, C), left ventricular posterior wall thickness at end diastole (LVPWd, D) and interventricular septum thickness at end diastole (IVSTd, E) of ND or HFD vehicle- or STA-treated mice.

F Quantification of the heart weight-to-body weight ratio (HW/BW).

G Quantification of myocyte cross-sectional area from heart cryosections.

H, I Expression levels of β-MHC (H) and BNP (I) were measured by qRT-PCR from cardiac tissues.

J Cardiac fibrosis was quantified on heart cryosections stained with Sirius red. Scale bar is 100 μm.

Data information: Data are presented as mean ± SEM. Student's *t*-test, *$P < 0.05$; **$P < 0.01$ ***$P < 0.001$ between indicated conditions, $n = 4$–8 mice per group.

activation of apoptosis (Fig 6C and E). Importantly, PIKfyve inhibition culminated in the reduction in mitochondrial $O_2^-$ levels (Fig 6A and B) in HFD-fed mice. As compared to vehicle-treated mice, myocardial levels of apoptosis (Fig 6C and D) and pro-apoptotic factor Bax (Fig 6E) were significantly lower in STA-treated HFD-fed mice. Electron microscopy analysis of mitochondrial integrity in cardiac tissue from HFD-fed mice revealed ultrastructural changes including decreased mitochondrial size and fragmented rounded interfibrillar mitochondria (Fig 7A and B), a typical hallmark of cardiac injury (Ong et al, 2010). In contrast, treatment of HFD-fed mice with STA preserved mitochondrial size and prevented mitochondrial damage as compared to vehicle-treated HFD-fed mice (Fig 7A and B). Defects in mitochondrial architecture are hallmarks for respiratory chain damage and ROS production. Therefore, we analyzed the expression profile of key complexes of the mitochondrial respiratory chain (OXPHOS complexes). Strikingly, in conditions of PIKfyve inhibition, we found an increased expression of mitochondrial-encoded genes in complexes I, II, III, IV, and V in cardiac tissue (Fig 7C and D) suggesting an improved respiratory efficiency.

### PIKfyve inactivation drives cardiac SIRT3 pathways in obesity-related cardiometabolic phenotype

We next studied the activation status of SIRT3 in STA-treated mice. It has been recently shown that phosphorylation of SIRT3 on

Ser/Thr residues led to increased enzymatic activity in mitochondria (Liu et al, 2015). Therefore, we performed immunoprecipitation of endogenous SIRT3 in heart extracts from control or STA-treated mice followed by immunoblot of phosphorylated proteins on serine residues. As shown in Fig 8A, STA treatment increased significantly the amount of phosphorylated SIRT3. Moreover, we investigated the acetylation status of two mitochondrial targets of SIRT3 involved in redox homeostasis, the superoxide dismutase 2 (SOD2) (Tao et al, 2010) and the isocitrate dehydrogenase 2 (IDH2) (Yu et al, 2012). STA treatment significantly reduced the amount of acetylated cardiac IDH2 (Fig 8B) and SOD2 (Fig 8C). Altogether, these data suggest that PIKfyve inhibition led to increased SIRT3 activity in hearts from obese mice.

### STA loses its anti-hypertrophic properties in SIRT3.KO mice

In order to confirm the implication of SIRT3 in the anti-hypertrophic effects of STA, we evaluated the effects of PIKfyve inhibition in SIRT3.KO mice treated with isoproterenol (ISO) to induce cardiac hypertrophy (Fig 9A and B). As shown in Fig 9B and C, treatment with STA significantly prevented ISO-induced cardiac hypertrophy as measured by decreased IVSTd and cardiomyocyte cross-sectional area as compared to vehicle-treated WT mice. In contrast, STA-mediated effects on cardiac hypertrophy were totally abrogated in SIRT3.KO mice (Fig 9B and C) suggesting that SIRT3 is required for the anti-hypertrophic activity of STA.

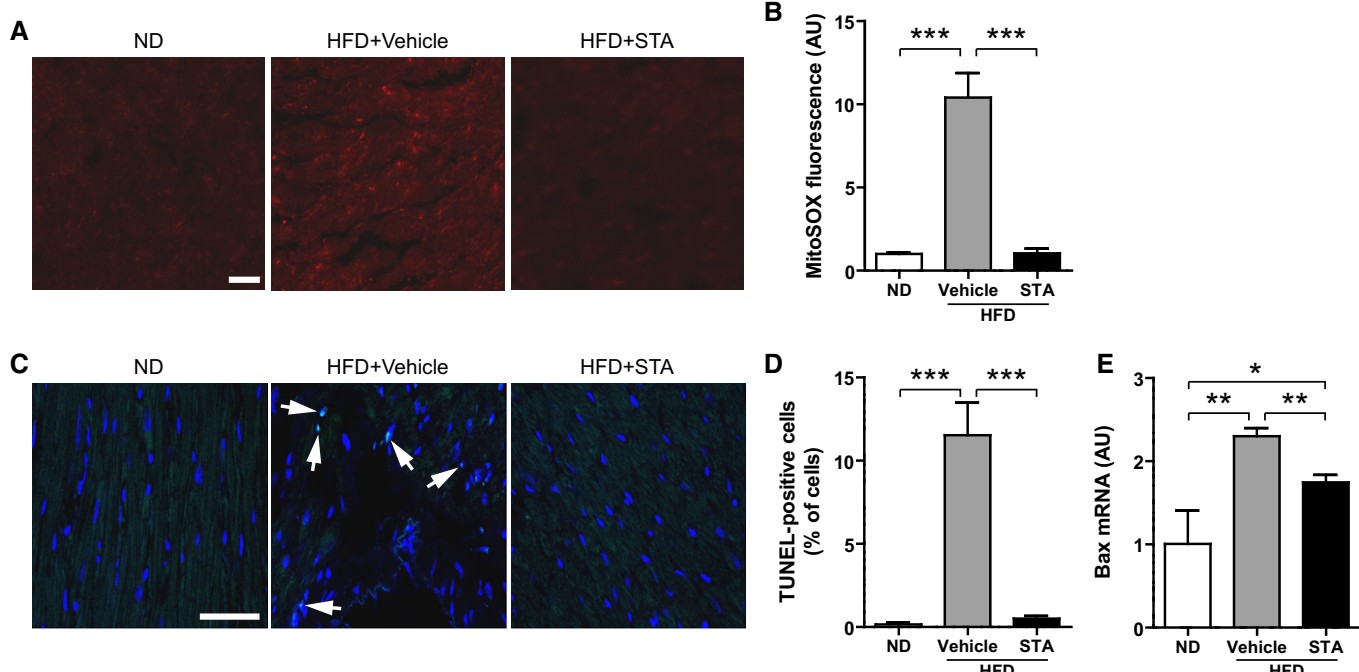

**Figure 6. Inhibition of PIKfyve decreases cardiac oxidative stress and apoptosis induced by obesity.**

A  Mitochondria-derived $O_2^-$ production was measured on heart cryosections using MitoSOX Red by confocal microscopy. Scale bar is 50 μm.
B  Quantification of MitoSOX fluorescence from (A).
C  TUNEL staining of heart cryosections showing apoptotic cells (arrows). The nuclei are stained in blue with DAPI. Scale bar is 50 μm.
D  Quantification from (C).
E  Bax expression level was measured by qRT-PCR from heart tissues.

Data information: Data are presented as mean ± SEM. Student's t-test, *P < 0.05; **P < 0.01 ***P < 0.001 between indicated conditions, n = 3–6 mice per group.

    

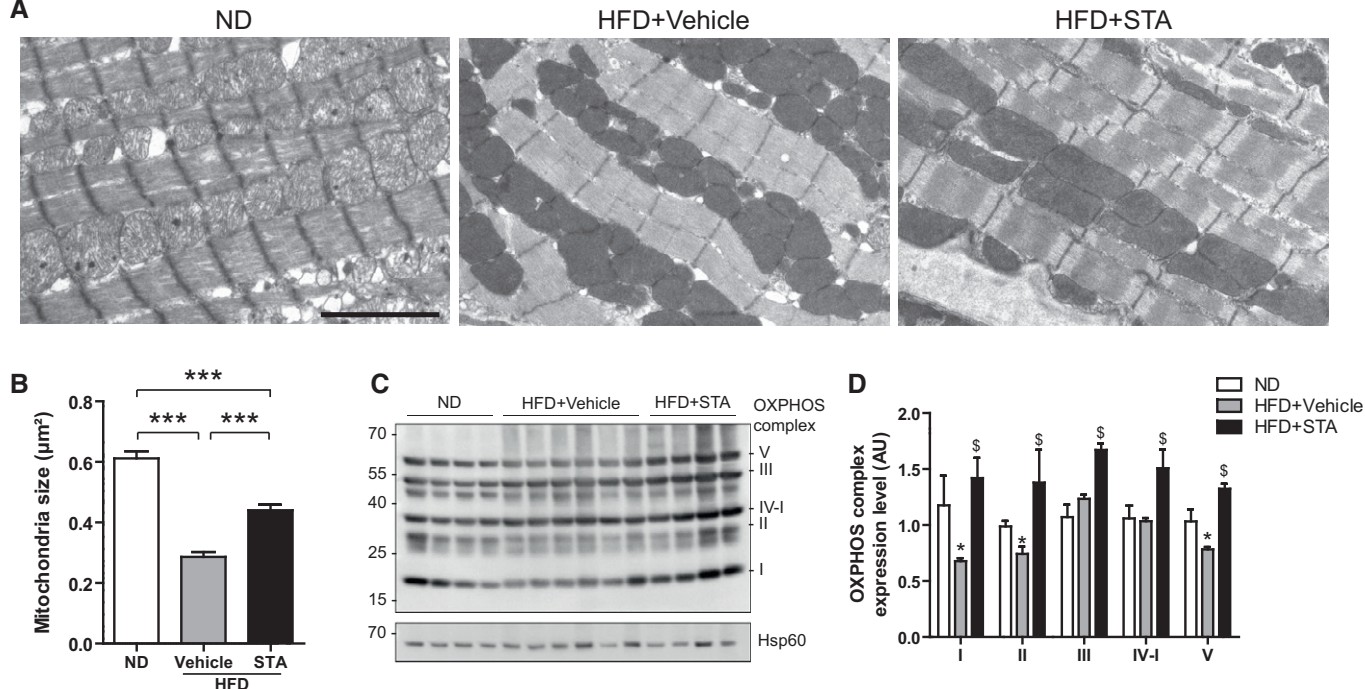

**Figure 7.   Chronic STA treatment reduces mitochondrial damages in obese mice.**

A   Electron micrographs showing preservation of myocardial mitochondrial structure in HFD STA-treated mice. Scale bar is 1 μm.
B   Quantification of mitochondria size from (A). Data are presented as mean ± SEM. Student's *t*-test, \*\*\**P* < 0.001 between indicated conditions, *n* = 4–6 mice per group.
C   Expression of several OXPHOS complexes was measured by Western blot on heart lysates.
D   Quantification of OXPHOS complexes expression from (C). Data are presented as mean ± SEM. Student's *t*-test, \**P* < 0.05 as compared with ND, $^{\$}P$ < 0.05 as compared with HFD+Vehicle. *n* = 3–6 mice per group.

Source data are available online for this figure.

Finally, in order to investigate whether PIKfyve inhibition affects cardiac SIRT3 protein content in mitochondria under hypertrophic stimulation, we examined the level of mitochondrial SIRT3 in control or ISO-treated mice hearts. As shown in Fig 9D, ISO treatment induced a reduction in mitochondrial level of SIRT3, most likely due to mitochondrial damage induced by ISO (Bloom & Cancilla, 1969). Strikingly, STA treatment induced a strong accumulation of SIRT3 in the mitochondria (Fig 9D). Taken together, these results indicate that PIKfyve inhibition reduces cardiac hypertrophy through mitochondrial SIRT3 activation.

## Discussion

Obesity is closely associated with cardiovascular and metabolic complications (Battiprolu *et al*, 2012). Increasing evidence suggests that abnormal mitochondrial ROS production and mitochondrial defects are at the center of the pathophysiology of the failing heart and metabolic disorders (Bournat & Brown, 2010; Tsutsui *et al*, 2011). The loss of mitochondrial integrity inevitably disturbs cell functions, sensitizes cells to stress and may trigger cell death, with potentially dramatic irreversible pathological consequences. In the present study, we unravel a critical role for the phosphoinositide kinase PIKfyve in the control of stress-induced mitochondrial

damage, ROS generation, apoptosis, and ventricular dysfunction in obesity-induced phenotype. Long-term HFD-induced obesity increases the heart workload, causes left ventricular hypertrophy, and impairs cardiac function (Battiprolu *et al*, 2012; Fuentes-Antras *et al*, 2015). Our *in vitro* results demonstrate that inhibition of PIKfyve attenuated stress-induced hypertrophic responses in cardiomyoblasts. In addition, in a mouse model of obesity-induced phenotype, we show that chronic treatment with STA decreased ventricular hypertrophy, a major predictor of cardiovascular events, and improves left ventricular contractility, suggesting a tight association between myocardial PIKfyve activity and cardiac function in the setting of obesity.

Obesity is associated with metabolic disorders leading to the installation of type 2 diabetes (Battiprolu *et al*, 2012). The present study is the first report that demonstrates the efficacy of pharmacological inhibition of PIKfyve on glycemic status in obesity-induced type 2 diabetes. If the total knockout of PIKfyve in mice is lethal at embryonic stage (Ikonomov *et al*, 2011), the generation of tissue-specific PIKfyve knockout mice has given some insights in the *in vivo* functions of the lipid kinase. Indeed, muscle-specific PIKfyve knockout mice are glucose intolerant and insulin resistant (Ikonomov *et al*, 2013). The key difference between the genetic and pharmacological inactivation of PIKfyve is that in KO mice, the protein is totally absent, preventing both the kinase activity and any

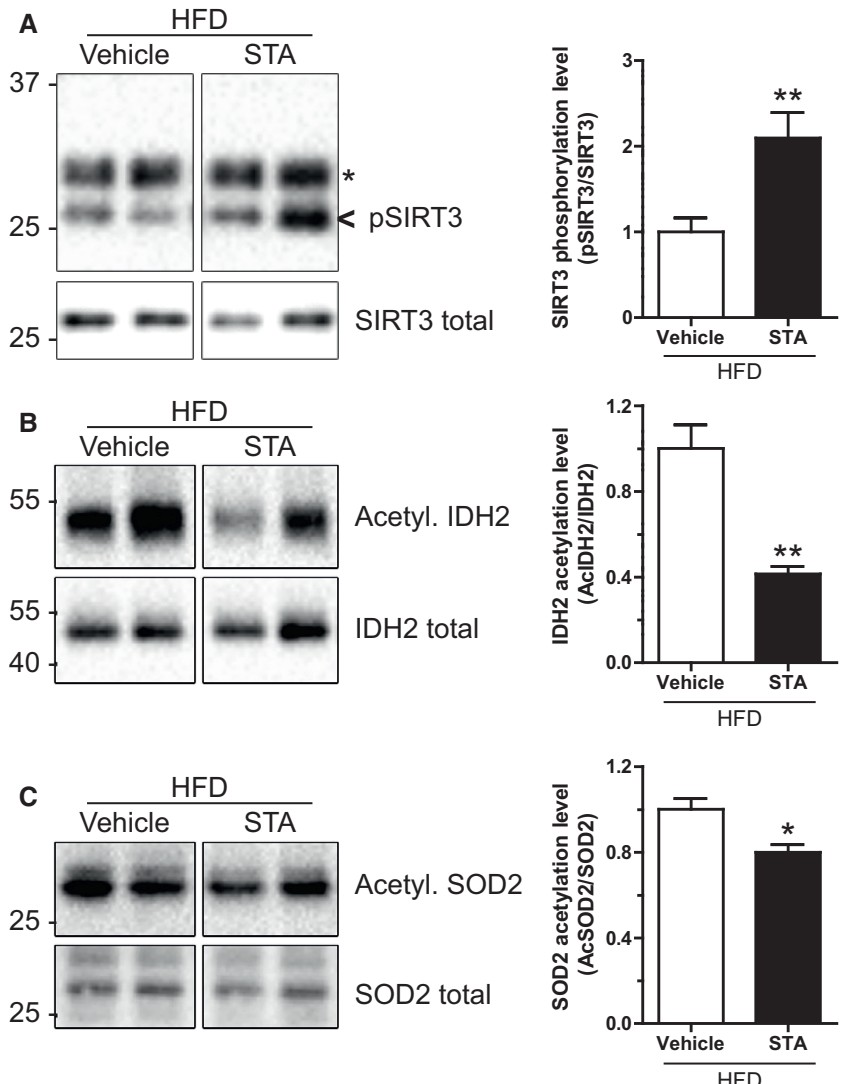

**Figure 8. SIRT3 is activated in hearts from STA-treated mice.**

A     Heart lysates from vehicle- or STA-treated obese mice were immunoprecipitated using an anti-SIRT3 antibody. Bound proteins were immunoblotted with an anti-phospho-serine antibody to reveal the phosphorylated SIRT3 (pSIRT3). As a loading control, total homogenates were probed with an anti-SIRT3 antibody (SIRT3 total). Asterisk indicates antibody light chains. Quantification is shown on the right panel.

B, C   Same as in (A) but lysates were immunoprecipitated with an anti-acetyl-lysine antibody and immunoblotted for IDH2 (B) and SOD2 (C). Quantifications are shown on the right panels.

Data information: Data are presented as mean ± SEM. Student's *t*-test, *$P < 0.05$; **$P < 0.01$, $n = 3$–8 mice per group.
Source data are available online for this figure.

scaffolding/docking function of PIKfyve. The use of pharmacological inhibitors allows a more detailed dissection of these two characteristics.

Our study provides the first evidence that PIKfyve controls the structural mitochondrial integrity and ROS production in cardiac cells through a SIRT3-dependent pathway. One might hypothesize that PIKfyve could be found on mitochondria to play such a role. However, PIKfyve is a cytosolic protein localized on endosomes through its FYVE domain (Sbrissa *et al*, 2002b). One might speculate that some lipid transfer may occur, allowing PIKfyve lipid products PI5P and/or PI(3,5)$P_2$ to be accumulated in mitochondrial

membranes. The capacity of phospholipids to alter membrane dynamics is widely recognized (van Meer *et al*, 2008), and interestingly, membrane fluidity has been shown to be a key factor in the respiratory chain efficiency (Waczulikova *et al*, 2007). Although a direct transfer between two distant lipid bilayers is very unlikely, there are examples of lipid exchange through protein carriers between two organelles, for example endoplasmic reticulum to Golgi apparatus (Moser von Filseck *et al*, 2015) or to plasma membrane (Stefan *et al*, 2013). Such a transfer mechanism could exist between the endosomal system and the mitochondria. Further work is needed to explore these possibilities and also to

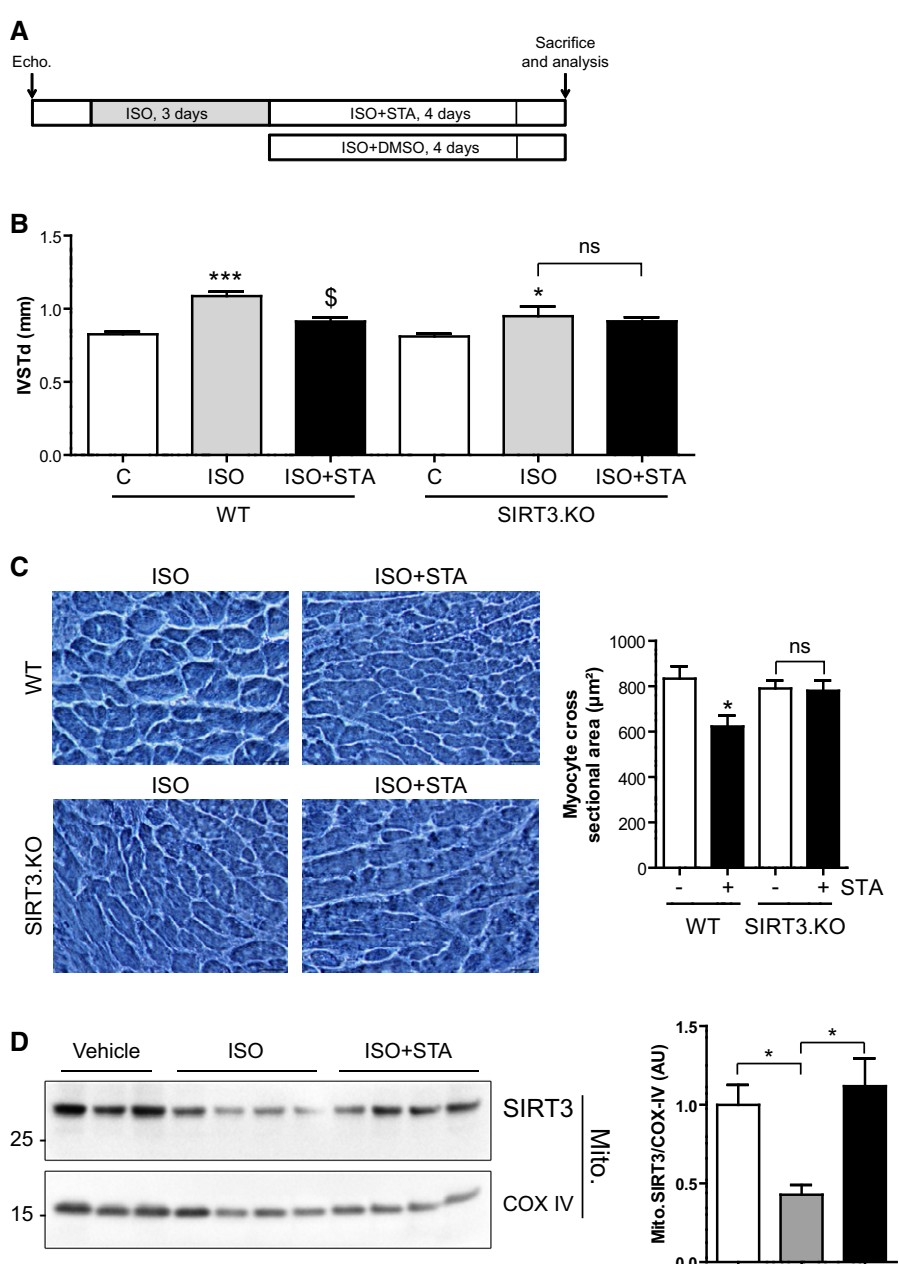

**Figure 9. STA loses its anti-hypertrophic properties in SIRT3.KO mice.**

A Schematic of the protocol used. WT or SIRT3.KO mice received isoproterenol (ISO) for 3 days followed by injections of ISO+STA or ISO+DMSO for 4 days. Echocardiography analysis was performed at the beginning and at the end of the protocol to compare the groups.

B IVSTd was measured in WT and SIRT3.KO mice treated with ISO or ISO+STA as indicated. Data are presented as mean ± SEM. Student's *t*-test, *$P < 0.05$; ***$P < 0.001$ as compared with control (C, before treatment) and $^\$P < 0.05$ as compared with ISO, from three to nine mice per group. ns, non-significant.

C Hematoxylin–eosin staining of heart cryosections from WT or SIRT3.KO mice after ISO or ISO+STA acute treatment and quantification of myocyte cross-sectional area. Scale bar is 25 μm. Data are presented as mean ± SEM. Student's *t*-test, *$P < 0.05$ as compared with ISO-treated group, from three to four mice per group. ns, non-significant.

D SIRT3 is enriched in mitochondria after STA treatment. Mitochondria were isolated from mice cardiac tissue treated as indicated, and blotted for SIRT3. COX IV was used as a loading control. Right panel shows quantification of mitochondrial SIRT3 normalized against COX IV. Data are presented as mean ± SEM. Student's *t*-test, *$P < 0.05$ between indicated conditions ($n = 3$–4 mice per group).

Source data are available online for this figure.

examine mitochondrial dynamics in the context of obesity-induced cardiomyopathy. Alternatively, we could not exclude that PIKfyve effects on mitochondria involve its protein kinase activity. Indeed,

it has been suggested that PIKfyve is able to phosphorylate several protein substrates (Ikonomov *et al*, 2003), although the regulation of this protein kinase activity remains poorly documented. It is

tempting to speculate that PIKfyve directly phosphorylates SIRT3 to control the redox status of the cell. However, given the fact that PIKfyve inhibition by STA increased SIRT3 phosphorylation and that only activating phosphorylations have been described for SIRT3 (Liu *et al*, 2015), one has to admit that PIKfyve controls SIRT3 activity through a different mechanism. One possibility would be that PIKfyve phosphorylates SIRT3 on a different site through an inhibitory phosphorylation. In any case, our results clearly demonstrate that PIKfyve is activated upon metabolic and oxidative stresses, adding new activation pathways to the already described osmotic shock (Sbrissa *et al*, 2002a). Taken together, these observations clearly identify PIKfyve as a stress sensor which is able to orchestrate the cell response.

A line of evidence suggests that obesity-induced cardiac dysfunction is linked to excessive mitochondrial ROS production, oxidative stress, and massive loss of cardiac cells (Sawyer *et al*, 2002; Battiprolu *et al*, 2012; Aurigemma *et al*, 2013). Both *in vitro* and *in vivo*, we show here that STA was able to reduce mitochondrial ROS generation, oxidative stress, and apoptosis. Interestingly, anti-hypertrophic activity of STA was associated with activation of SIRT3 pathways in the heart. SIRT3 has been shown to be a negative regulator of cardiac hypertrophy (Sundaresan *et al*, 2009), ROS production (Qiu *et al*, 2010), apoptotic cell death (Sundaresan *et al*, 2008), metabolism (Alfarano *et al*, 2014), and aging (McDonnell *et al*, 2015). Our data suggest the existence of a novel PIKfyve-dependent regulatory pathway that impinges on key processes of mitochondrial biogenesis. This is supported by the following evidences: First, PIKfyve inhibition increases myocardial phosphorylation level of SIRT3; second, PIKfyve inhibition decreases mitochondrial ROS production; third, PIKfyve inhibition is able to increase OXPHOS components in cardiac tissue; and fourth, PIKfyve inactivation causes an increase in mitochondrial size in the heart.

Despite its fundamental role, little is known on how SIRT3 is translocated into the mitochondrial matrix (Schwer *et al*, 2002). Here, we show that PIKfyve inhibition induced the translocation of SIRT3 to the mitochondria, independently of stress. We postulate that such a regulation would prime the cell for a quick response under stress and ultimately protecting it from ROS overproduction and cell death. In that regard, deciphering the molecular mechanisms involved in STA-dependent SIRT3 translocation and activation would help to better understand the regulation of SIRT3, and would pave the way to new therapies for diseases associated with SIRT3 deficiency.

With the increasing prevalence of obesity-associated metabolic and cardiovascular disorders, our study places the lipid kinase PIKfyve in the context of cardiometabolic diseases and highlights the therapeutic potential of PIKfyve pharmacological inhibition to limit mitochondrial damage and to improve cardiometabolic phenotype in obese patients.

# Materials and Methods

### Reagents and antibodies

Antibodies used in this study are the following: anti-GAPDH (sc-32233), anti-HSP90 (sc-13119), anti-Drp1 (H-300), and anti-caspase 3 (sc-7148) from SantaCruz Biotechnology; anti-phospho-serine

(4A4) from Millipore; anti-OXPHOS/COX (MS604/G2594) from Mitosciences; and anti-acetylated-lysine (9441S), anti-SIRT3 (D22A3), anti-SIRT1 (1F3), anti-COX IV (4844), and anti-cleaved caspase 3 (9661) from Cell Signaling Technology. All antibodies were used at 1:1,000 for immunoblot and 1:100 for immunofluorescence. Fluorescent Alexa-coupled secondary antibodies (used at 1:300) and DAPI were from Life Technologies. HRP-coupled secondary antibodies (used at 1:3,000) were from Cell Signaling Technology. STA-5326 was purchased from Axon MedChem and was referred to as STA throughout this study. All other chemicals were from Sigma-Aldrich unless otherwise stated.

### Molecular biology

siRNA against SIRT3 were from Eurogentec and as follows: 5′-GCGU UGUGAAACCCGACAU-3′ and 5′-AUGUCGGGUUUCACAACGC-3′. siRNA Universal negative control was from Sigma. siRNA against PIKfyve were from Sigma and as follows: 5′-GUUGUCAAUGGCUUU GUUU-3′ and 5′-AAACAAAGCCAUUGACAAC-3′. Primers for qRT-PCR used in this study are as detailed in Table EV1.

### Quantitative RT–PCR analysis

Total RNAs were isolated from cultured mouse cardiac fibroblasts using the RNeasy mini kit (Qiagen). Total RNAs (300 ng) were reverse transcribed using Superscript II reverse transcriptase (Invitrogen) in the presence of a random hexamers. Real-time quantitative PCR was performed as previously described (Alfarano *et al*, 2014). The expression of target mRNA was normalized to GAPDH mRNA expression.

### Evaluation of apoptosis and ROS production

Apoptosis level was assessed as before (Boal *et al*, 2016) using the DeadEnd Fluorometric TUNEL system according to manufacturer's instructions (Promega). Mitochondrial $O_2^-$ and $H_2O_2$ were measured using MitoSOX Red indicator (Life Technologies) and MitoPY1 (Sigma-Aldrich) as described (Boal *et al*, 2015b). Mitochondrial superoxide levels on heart cryosections were assessed as described elsewhere (Sun *et al*, 2013).

### Quantification of PIKfyve product PI5P

Quantification of the PI5P level was performed using a mass assay as described before (Morris *et al*, 2000) with slight modifications (Dupuis-Coronas *et al*, 2011).

### Animal studies, experimental protocol, and metabolic measurements

The investigation conforms to the Guide for the Care and Use of Laboratory Animals published by the US National Institutes of Health (NIH Publication No. 85-23, revised 1985) and was performed in accordance with the recommendations of the French Accreditation of the Laboratory Animal Care (approved by the local Centre National de la Recherche Scientifique ethics committee). Two-month-old wild-type male C57BL6/J mice purchased from Janvier Labs were fed a high fat diet (HFD, 45% fat) for 12 months.

Animals were randomly divided into two groups ($n = 8$ each). Mice then received for 17 consecutive days intraperitoneal injections of STA (2 mg/kg/day) or vehicle (DMSO), corresponding to a final DMSO concentration of 50% diluted in PBS. The dose of STA was selected on the basis of our preliminary animal studies. The efficiency of PIKfyve inhibition was monitored by the quantification of cardiac PI5P (Fig EV4A). Plasma glucose (Accu-check, Roche Diagnostics) was measured in fasted state. LPO (lipid peroxide) quantification was done as described before (Foussal *et al*, 2010) using an ELISA-based kit (Cayman). Triglycerides were quantified using enzymatic assay (TG enzymatic PAP150, Biomerieux). Plasma insulin was measured using an ELISA-based kit (Mercodia). Plasma IL-6 and TNF-α were quantified using ELISA kit (eBiosciences). Mice genetically invalidated for SIRT3 and their controls have been described previously and display no significant phenotype under basal conditions (Bochaton *et al*, 2015). In order to induce cardiac hypertrophy, WT or SIRT3.KO mice ($n = 9$ per group) were intraperitoneally treated with ISO (15 mg/kg/day) for 3 days. Mice were then randomly segregated into two groups and treated for 4 days with either ISO+DMSO or ISO+STA (2 mg/kg/day).

### Echocardiographic studies

Blinded echocardiography was performed as described (Pchejetski *et al*, 2012) on isoflurane-anesthetized mice using a Vivid7 imaging system (General Electric Healthcare) equipped with a 14-MHz sectorial probe. Two-dimensional images were recorded in parasternal long- and short-axis projections, with guided M-mode recordings at the midventricular level in both views. Left ventricular (LV) dimensions and wall thickness were measured in at least five beats from each projection and averaged. Fractional shortening and ejection fraction were calculated from the two-dimensional images.

### Morphology

Ultrastructural studies of cardiac tissue by electron microscopy were done as before (Boal *et al*, 2015b). Briefly, cardiac tissues were fixed in cold 2.5% glutaraldehyde/1% paraformaldehyde, post-fixed in 2% osmium tetroxide, embedded in resin, and sectioned. Hematoxylin–eosin and Sirius red stainings of heart cryosections were done according to standard methods. The extent of cardiac fibrosis was quantified using ImageJ software (Pchejetski *et al*, 2012). Quantification of myocyte cross-sectional area was performed as described (Foussal *et al*, 2010).

### Cell culture, transfection, and treatments

The rat embryonic cardiomyoblastic cell line H9C2 (ATCC) was cultured in DMEM medium (Life Technologies) supplemented with 10% FBS and 1% penicillin–streptomycin in a 37°C, 5% $CO_2$ incubator. siRNA transfection was performed with Lipofectamine RNAiMAX (Life Technologies) according to manufacturer's instructions. For hypoxic treatment, cells were pretreated for 30 min with STA (100 nM) or DMSO (vehicle only) and then subjected to normoxia (5% $CO_2$; 21% $O_2$, balance $N_2$) or hypoxia in a hypoxic chamber (5% $CO_2$, 1% $O_2$, balance $N_2$) for 2 h (for ROS measurement) or 16 h (for apoptosis). To assess cell hypertrophy, the medium was replaced and cells were further incubated for 24 h in normoxic conditions (reoxygenation) in the continuous presence of STA or DMSO. To induce metabolic stress, the cells were treated with 2-deoxy-D-glucose (2DG, 50 mM) in complete medium for 4 h (for ROS production) or 24 h for apoptosis. Cell viability was assessed using MTT colorimetric assay.

### Immunofluorescence, determination of mitochondrial fragmentation, and cell size measurement

Immunofluorescence was performed as previously described (Boal *et al*, 2015a). For determination of mitochondrial fragmentation, H9C2 cells were live-stained with MitoTracker Red CMXRos (Life Technologies) at 200 nM for 15 min at 37°C. Fixed cells were then imaged by widefield microscopy. Mitochondrial fragmentation was measured using a dedicated ImageJ plugin as described (Dagda *et al*, 2009). For cell size measurement, three fields of view were randomly selected per conditions and cell surface was quantified using ImageJ.

### Isolation of mitochondria from H9C2 cells

Cells were extensively washed in ice-cold PBS, scraped in HB (20 mM HEPES, 270 mM sucrose, protease/phosphatase inhibitor cocktails from Biotools, pH 7.4), and disrupted by 10 strokes through a 27-gauge needle. Cell debris were pelleted by centrifugation (10 min 400 *g* 4°C). The supernatant was further spun 25 min 10,000 *g* 4°C. The resulting pellet corresponding to the isolated mitochondria was solubilized in RIPA buffer (50 mM Tris–HCl (pH 7.5), 150 mM NaCl, 0.1% SDS, 0.5% DOC, 1% Triton X-100, 1 mM EDTA, and protease/phosphatase inhibitor cocktails).

### Isolation of mitochondria from mice heart

Mitochondria were isolated from mouse hearts essentially as described (Fazal *et al*, 2017).

### Protein extraction, immunoprecipitation, and Western blotting

Proteins from cardiac tissues and H9C2 cells were extracted using RIPA buffer and quantified using the Bio-Rad Protein Assay (Bio-Rad). For immunoprecipitation, lysates (500 μg proteins) were incubated overnight at 4°C with 5 μl of the specific antibody bound to Protein G Sepharose 4 Fast Flow (GE Healthcare). After extensive washes, bound proteins were eluted in Laemmli sample buffer (50 mM Tris–HCl (pH 6.8), 2% SDS, 6% glycerol, 0.2 mM DTT, and 0.02% bromophenol blue) and denaturated at 70°C for 15 min. Proteins were resolved by SDS–PAGE and Western blotting. Immunoreactive bands were detected by chemiluminescence with the Clarity Western ECL Substrate (Bio-Rad) on a ChemiDoc MP Acquisition system (Bio-Rad).

### Statistical analysis

Data are expressed as mean ± SEM. Statistical comparison between two groups was performed by Student's *t*-test, while comparison of multiple groups was performed by one- or two-way ANOVA followed by a Bonferroni's *post hoc* test when appropriate using GraphPad Prism version 5.00 (GraphPad Software, Inc).

## The paper explained

### Problem
Obesity is a growing health issue in modern societies and is associated with cardiovascular diseases, metabolic complication, and cardiac dysfunction. Increasing evidences suggest that mitochondrial defects are central to the pathophysiology of the failing heart and metabolic disorders. However, the specific mechanisms remain unclear, and the clinical management of this multifactorial disease requires a multidisciplinary approach and new therapeutic strategies.

### Results
Here, we show that the lipid kinase PIKfyve is critical for stress-induced mitochondrial damage in myocardium. Pharmacological inhibition of the kinase by STA-5326 reduces mitochondrial ROS production and apoptosis through the deacetylase SIRT3. Moreover, chronic PIKfyve inhibition reduces ventricular hypertrophy and improves cardiac function in morbidly obese mice.

### Impact
Our study reveals the importance of PIKfyve inhibition in the treatment of cardiomyopathy in obese mice and propose new therapeutic avenue to improve cardiometabolic phenotype in obese patients.

**Expanded View** for this article is available online.

## Acknowledgements
This project was supported by grants from Région Midi-Pyrénées (HT, MC, and FB) and Fondation Lefoulon-Delalande (FB), MEDEA Erasmus Mundus Program (MC and AT). We are grateful to Sophie Legonidec and Anexplo platform for mice phenotyping. We thank Claire Vinel, Gwendoline Astre, and Simon Deleruyelle for technical assistance. We are grateful to Jean-Emanuel Sarry (CRCT, Toulouse) for the anti-SOD2 and IDH2 antibodies and Hélène Authier (IRD, Toulouse) for IL-12 and IL-23 primers. We thank Franck Desmoulin for his help in statistical analysis and Olivier Lairez for echocardiography analysis.

## Author contributions
FB, OK, and HT conceived and designed the experiments and analyzed the data. FB, MC, AT, LG, HT, and OK performed the experiments. HTh, CV, and DB performed the experiments on the SIRT3.KO mice. BP, PV, and AP assisted in the development of the project. FB, HT, and OK co-wrote the manuscript.

## Conflict of interest
The authors declare that they have no conflict of interest.

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
