## [Review Process File · EMBO Molecular Medicine]

Inhibition of PIKfyve prevents myocardial apoptosis and hypertrophy through activation of SIRT3 in obese mice

Helene Tronchere, Mathieu Cinato, Andrei Timotin, Laurie Guitou, Camille Villedieu, Helene Thibault, Delphine Baetz, Bernard Payrastre, Philippe Valet, Angelo Parini, Oksana Kunduzova and Frederic Boal

Corresponding author: Frederic Boal, INSERM I2MC U1048

Review timeline:

Submission date:	21 September 2016
Editorial Decision:	31 October 2016
Revision received:	17 February 2017
Editorial Decision:	07 March 2017
Revision received:	14 March 2017
Accepted:	15 March 2017

Transaction Report:

Editor: Céline Carret

1st Editorial Decision

31 October 2016

Thank you for the submission of your manuscript to EMBO Molecular Medicine and for your patience. We have now heard back from the three referees whom we asked to evaluate your manuscript.

Although the referees find the study to be of potential interest, they also raise a number of concerns about the conclusiveness of the results, and several technical issues. While referee 1 is rather positive and enthusiastic about the data, referees 2 and 3 are more reserved and highlight overlapping concerns about technical limitations regarding the blots, normal diet controls etc. Furthermore, and importantly for the scope of our journal, referee 3 highlights serious limitations of the experimental design and provide suggestions to improve the study in terms of mechanistic understanding and conclusiveness.

Given these comments, we would be willing to consider a revised manuscript with the understanding that the referee concerns must be fully addressed (and experimentally as much as possible) and that acceptance of the manuscript would entail a second round of review.

I look forward to seeing a revised form of your manuscript as soon as possible.

***** Reviewer's comments *****

Referee #1 (Remarks):

In this study, Tronchere et al show that PIKfyve controls \ mitochondrial fragmentation, hypertrophic and apoptotic responses to stress, in part by activating SIRT3. A combination of cell and animal models show that STA treatment suppresses excessive mitochondrial ROS production and apoptosis and improves cardiometabolic profile in a mouse model of cardiomyopathy linked to obesity. Overall, this study is well executed and the authors should be commended for experimental rigor (multiple models, pharmacological and genetic manipulations). I only offer minor comments for the authors to consider.

Minor:

-SIRT3 has been shown to influence mitochondrial fission/fusion; does knock-down of SIRT3 in cells alter the mitochondrial network in response to STA?

-Both SIRT3 as well as all 5 OXPHOS components were elevated upon STA treatment. Have the authors considered whether the increases in SIRT3 are part of global mitochondrial expansion/biogenesis phenotype?

-Have the authors attempted to measure NAD⁺ or NADH in response to cellular/metabolic stress +/- STA treatment? If NAD⁺ levels change, this might provide insight into how SIRT3 activity changes in their system.

Referee #2 (Remarks):

The authors present data on STA-mediated ameliorative effects on cardiac hypertrophy with an angle towards the utility of this drug to prevent obesity associated cardiac hypertrophy. To accomplish this, the authors use a combination of cell and animal models to understand the protective effects of STA, a PIKfyve inhibitor. This manuscript is potentially important and could result in a novel therapeutic avenue but I have several questions that would help in interpreting these data.

Major points

- The authors perform several mechanistic experiments in cultured cardiomyocyte cells using hypoxia and 2DG to induce cell stress. It is unclear how similar these effects are to obesity-induced hypertrophy and the authors should be cautious in relating these two models. As an example, the role of SIRT3 was not tested in obesity in this paper, in contrast to the implications of the title, which could be reworded as there may be other mechanisms linking STA to cardiometabolism in vivo
- The specificity of STA is not clearly described, and could be affecting other processes in vivo or in vitro. Specificity of STA could be more strongly established by showing no further effect after PIKfyve knockdown
- The authors show efficacy of STA via an in vitro mass assay as opposed to HPLC mediated separation of phosphatidylinositides. Details are lacking regarding this method, as the reference to Pendairies et al describes a HPLC assay not the assay used in the paper (itself refers to an earlier paper). To this point, it is unclear, if STA is effective, why PI(5)P is unaffected in the absence of hypoxia. Furthermore, as PIKfyve alters both PI(5)P and PI(3,5)P₂ it is important to measure the effects of hypoxia, 2DG and STA treatment on both lipids (again, ideally by stable isotope HPLC methods)
- The mechanism by which SIRT3 mitochondrial levels are induced in vitro is not explored in the in vivo model, rather phosphorylation of SIRT3 is described for the animal models. Please provide mitochondrial SIRT3 levels for the in vivo model, and phosphorylation of SIRT3 and acetylation of IDH2/SOD2 in the in vitro model. This would allow more clear assessment of the animal and cell culture models and how concordant the mechanisms are
- Please provide data on the normal diet animals in Figures 6-8, as it is important whether the STA treated groups are more similar to ND or not

- For the mouse experiments, showing body composition for the HFD/STA group would go a long way to understanding the physiological changes caused by STA

Minor points

- Is the upper the upper band in panel 4C nonspecific? If so, could you please provide evidence and mark it as such with an asterisk.
 - Two way ANOVA's testing for interactions between STA and stress (or STA/stress/knockdown) may be more appropriate to evaluate these data.

Referee #3 (Remarks):

In this manuscript, the authors described a lipid kinase PIKfyve as a regulator of cardiometabolic status and mitochondrial integrity using a diet-induced obesity mouse model. In vitro and in vivo experiments were employed to demonstrate the effect of PIKfyve inhibition on the reverse of obesity -induced cardiac mitochondrial damage and apoptosis potentially through SIRT3 signaling. While the results provided some insights into the potential mechanisms related to obesity-associated myocardial abnormalities, for the most part, the experiments described only associations among some potential pathways. The manuscript has the following major limitations that the authors should consider for future submission:

- 1) The limitation on the obesity model: while most of the diet-induced obesity model using a 45% high fat diet only need 5 months for the development of obesity, why did the authors choose to feed the mice for 12 months which is a very long period compared to most of the published obesity models? How do the authors distinguish the effects between obesity and aging?
- 2) The authors totally ignored the diastolic dysfunction associated with high fat diet feeding. This is a major flaw of the experimental design, as diabetes is first associated with diastolic abnormalities.
- 3) As PIKfyve is ubiquitously expressed, in which cell type the inhibition with STA affords the protection? In order to determine any effects related to PIKfyve, data on the expression of PIKfyve should be presented.
- 4) The high fat diet-induced obesity is a chronic progression, especially in the current manuscript, how could the acute treatment with STA for 17 days have such a long-lasting chronic effect to reverse all the effects associated with obesity?
- 5) There are a number of serious issues related to the techniques used including IHC staining, western blotting, echo, ...
 - a. The IHC staining and related conclusions are not convincing at all. The image resolutions are very poor and any conclusions from those images are doubtful.
 - b. The authors need to provide the 2-D M-mode images as the frequency used is relatively low and a convincing resolution should be presented.
 - c. The western blots need appropriate ND controls; and most importantly the proteins should be blotted on the same membrane and displayed so. The authors also should provide a rationale on using IP to detect the phosphor-SIRT3.
 - d. The cytokines concentration should be provided.

1st Revision - authors' response

17 February 2017

Referee #1 (Remarks):

In this study, Tronchere et al show that PIKfyve controls \ mitochondrial fragmentation, hypertrophic and apoptotic responses to stress, in part by activating SIRT3. A combination of cell and animal models show that STA treatment suppresses excessive mitochondrial ROS production and apoptosis and improves cardiometabolic profile in a mouse model of cardiomyopathy linked to obesity. Overall, this study is well executed and the authors should be commended for experimental rigor (multiple models, pharmacological and genetic manipulations). I only offer minor comments for the authors to consider.

Minor:

-SIRT3 has been shown to influence mitochondrial fission/fusion; does knock-down of SIRT3 in cells alter the mitochondrial network in response to STA?

We thank the reviewer for his helpful comments and careful reading of our manuscript.

The majority of studies have been performed on in vivo models demonstrating that SIRT3 deficiency in mice is linked to impaired mitochondria function, structure and elevated ROS levels (Kim et al, 2010; Samant et al, 2014). It has been suggested that SIRT3-dependent changes in mitochondrial function were linked to increased susceptibility to oxidative stress with age (Kim et al, 2010; McDonnell et al, 2015). In order to test whether silencing of SIRT3 could affect mitochondria structure, we performed additional experiments on H9C2 cells silenced for SIRT3 expression using siRNA. As shown below, silencing of SIRT3 did not alter overall mitochondria morphology in H9C2 cardiomyoblasts, alone or in response to STA. Our results suggest that in basal state endogenous SIRT3 is not required to maintain mitochondrial integrity in H9C2 rat cardiomyoblasts, and could reflect some cell-type specificity in its function. We do not feel necessary to add these data to the manuscript but if the reviewer thinks otherwise we are willing to do so.

SIRT3 silencing does not alter mitochondrial structure in H9C2 cells. H9C2 cells were transfected with a non-targeting siRNA (siControl) or a siRNA against SIRT3 (siSIRT3), treated or not with STA, fixed and stained for Hsp60 to label the mitochondria. Bar is 10 μ m. Right panel shows quantification of the mitochondrial fragmentation index. ns, non-significant.

-Both SIRT3 as well as all 5 OXPHOS components were elevated upon STA treatment. Have the authors considered whether the increases in SIRT3 are part of global mitochondrial expansion/biogenesis phenotype?

We thank the reviewer for this suggestion. SIRT3 has been shown to orchestrate fundamental aspects of mitochondrial biogenesis and function by regulating ATP generation from oxidative phosphorylation (Weir et al, 2013). Remarkably, the basal levels of ATP in the heart of SIRT3 null mice are reduced by more than 50%, suggesting that SIRT3 plays an important role in maintaining cardiac energy homeostasis (Ahn et al, 2008). SIRT3 is central in the maintenance of mitochondrial performance not only by regulating energy homeostasis but also, as recent studies have shown, by limiting oxidative stress (Bell & Guarente, 2011). Our data suggest the existence of a novel PIKfyve-dependent regulatory pathway that impinges on key processes of mitochondrial biogenesis. This is supported by the following evidences: first, PIKfyve inhibition increases myocardial phosphorylation level of SIRT3, a recently described activator mechanism (Liu et al, 2015); second, PIKfyve inhibition decreases mitochondrial ROS production. Interestingly, we found that siRNA-mediated knockdown of SIRT3 abolishes the effects of STA on mitochondrial ROS level, leading us to conclude that PIKfyve induces mitochondrial ROS production through a SIRT3-dependent pathway. Third, PIKfyve inhibition is able to increase myocardial expression of key complexes of the mitochondrial respiratory chain (OXPHOS components). Fourth, PIKfyve inactivation causes an increase in mitochondrial size in the heart. The discussion section has been modified accordingly.

-Have the authors attempted to measure NAD⁺ or NADH in response to cellular/metabolic stress +/- STA treatment? If NAD⁺ levels change, this might provide insight into how SIRT3 activity changes in their system.

The hypothesis that PIKfyve inhibition could control NAD⁺ levels in cells, therefore regulating SIRT3 activity is very interesting and we thank the reviewer for this suggestion. We have performed

additional experiments to quantify NAD⁺ levels in H9C2 cells treated or not with STA and submitted to hypoxia for 2h. Across 3-4 independent experiments we could not find any modifications in the levels of NAD⁺ in treated cells (either in response to STA or hypoxia). These data, summarized below, are consistent with the fact that we show that SIRT1, a closely-related NAD⁺-dependent deacetylase but nuclear isoform, is apparently not regulated by PIKfyve inhibition (see Figure 3E-F). We do not feel necessary to add these data to the main manuscript but are willing to do so if the reviewer thinks otherwise.

N	N+STA	H	H+STA
1.99+/-0.43	2.29+/-0.55	2.34+/-0.56	1.38+/-0.23

H9C2 cells were treated as indicated for 2h, lysed and NAD⁺ was quantified by fluorescence method. Results are presented as mean+/-SEM from quadruplicate experiment.

Referee #2 (Remarks):

The authors present data on STA-mediated ameliorative effects on cardiac hypertrophy with an angle towards the utility of this drug to prevent obesity associated cardiac hypertrophy. To accomplish this the authors use a combination of cell and animal models to understand the protective effects of STA, a PIKfyve inhibitor. This manuscript is potentially important and could result in a novel therapeutic avenue but I have several questions that would help in interpreting these data.

Major points

- The authors perform several mechanistic experiments in cultured cardiomyocyte cells using hypoxia and 2DG to induce cell stress. It is unclear how similar these effects are to obesity-induced hypertrophy and the authors should be cautious in relating these two models. As an example, the role of SIRT3 was not tested in obesity in this paper, in contrast to the implications of the title, which could be reworded as there may be other mechanisms linking STA to cardiometabolism in vivo

We are very grateful to the reviewer for his comments. Oxidative and metabolic stresses are key factors in the pathogenesis of obesity-related diseases (Bournat & Brown, 2010; Tsutsui et al, 2011). To provide mechanistic insight into obesity-induced cardiac injury, we performed in vitro experiments in cardiomyoblasts subjected to hypoxia-induced oxidative stress or 2-deoxy-D-glucose (2DG)-induced metabolic stress. In order to clarify this point, in the new version of the manuscript, we have stated the relevance of these cellular models in the context of obesity-induced diseases.

Regarding the implication of SIRT3 in vivo: we now provide compelling evidence regarding the implication of SIRT3 in the anti-hypertrophic effects of STA using SIRT3 KO mice. Our data demonstrate that anti-hypertrophic activity of STA is totally abrogated in SIRT3 KO mice (Figure 9B-C). In addition, in the new version of our manuscript we show that STA increases SIRT3 protein level in isolated mitochondria from cardiac tissue (Figure 9D). Taken together, these results suggest that PIKfyve inhibition promotes mitochondrial SIRT3 activation under hypertrophic stimulation in vivo.

We believe that these new data greatly strengthen our study. Accordingly, we have reworded the title.

- The specificity of STA is not clearly described, and could be affecting other processes in vivo or in vitro. Specificity of STA could be more strongly established by showing no further effect after PIKfyve knockdown

Thanks for this point. Recently, STA specificity has been clearly described by Cai and colleagues (Cai et al, 2014). In order to confirm PIKfyve involvement, we now provide additional data on H9C2 cells depleted for endogenous PIKfyve by siRNA on ROS production, mitochondria fragmentation and Drp1 oligomerization (see figures EV1C-E). Furthermore, we now show in figure EV1C that in cells depleted for PIKfyve, STA has no further effect on hypoxia-induced ROS production, validating PIKfyve as the target for the anti-oxidant properties of STA.

- The authors show efficacy of STA via an in vitro mass assay as opposed to HPLC mediated separation of phosphatidylinositides. Details are lacking regarding this method, as the reference to Pendairies et al describes a HPLC assay not the assay used in the paper (itself refers to an earlier paper).

Although the Pendairies et al paper also described PI5P quantification by mass assay, we agree with the reviewer that details are lacking in this publication. We have now added the original publication that has first described the PI5P mass assay (Morris et al, 2000) and rephrased the material and methods section as follow: "Quantification of the PI5P level was performed using a mass assay as described (Morris et al, 2000) with slight modifications (Dupuis-Coronas et al, 2011)."

To this point, it is unclear, if STA is effective, why PI(5)P is unaffected in the absence of hypoxia.

Indeed the basal level of PI5P is not altered by STA treatment in H9C2 cells (Figure 1A-B), indicating that PI5P basal turn-over in cardiomyoblasts is maintained by a PIKfyve-independent pathway. Accordingly, in the new version of our manuscript we have modified the results section.

Furthermore, as PIKfyve alters both PI(5)P and PI(3,5)P₂ it is important to measure the effects of hypoxia, 2DG and STA treatment on both lipids (again, ideally by stable isotope HPLC methods).

PIKfyve is indeed responsible for the synthesis of both lipids. However, PI(3,5)P₂ is present in very low amount in cells (typically 0.01% of total phosphoinositides (Ikononov et al, 2006; Kim et al, 2016; Zolov et al, 2012), therefore its quantification is extremely difficult by the metabolic and HPLC method. Quantification of PI5P by HPLC is also quite challenging since this lipid has a tendency to co-migrate with PI4P and typically represents only 1-2% of PI4P levels (Sarkes and Rameh, 2010). Following infection by the bacteria *S. flexneri*, mammalian cells produce a very high amount of PI5P which can then be measured by the metabolic and HPLC method (Niebuhr et al, 2002) but except this peculiar condition it is difficult to measure PI5P by HPLC in mammalian cells even by using a tandem of HPLC column (Sarkes and Rameh, 2010). In this regard, we have purposely used the mass assay because it allows the quantification of low levels PI5P. Moreover, the mass assay allows the quantification of PI5P in tissue (see PI5P quantification in mice heart in Fig EV3A), which is impossible with the metabolic labeling method. Unfortunately, a mass assay to quantify PI(3,5)P₂ does not exist yet and this lipid clearly remains the most difficult phosphoinositide to study in mammalian cells.

In our study, PI5P quantification was mainly used as a readout for PIKfyve activity and to validate the efficacy of STA treatment in our in vitro and in vivo models. We agree with the reviewer that measuring PI(3,5)P₂ levels would be of great interest, particularly in order to test the implication of both lipid products in the observed effects of STA. In this context, we have modified our discussion which was entirely focused on PI5P, taking into account a potential role for both lipids.

- The mechanism by which SIRT3 mitochondrial levels are induced in vitro is not explored in the in vivo model, rather phosphorylation of SIRT3 is described for the animal models. Please provide mitochondrial SIRT3 levels for the in vivo model, and phosphorylation of SIRT3 and acetylation of IDH2/SOD2 in the in vitro model. This would allow more clear assessment of the animal and cell culture models and how concordant the mechanisms are.

Thanks for this direction. In the new version of our manuscript, we explore the potential mechanism in the in vivo model. Using an isoproterenol-induced model of cardiac hypertrophy in mice, we show that STA-dependent anti-hypertrophic effect is totally abrogated in SIRT3 KO mice (Fig.9B-C). In addition, we demonstrate that STA increases SIRT3 protein levels in isolated cardiac mitochondria in response to hypertrophic stimuli (Fig.9D). Taken together, these results provide the first evidence that PIKfyve inhibition promotes mitochondrial SIRT3 activation under hypertrophic stimulation in vivo. We have revised our paper accordingly and feel that your comments helped clarify and improve our paper.

Regarding the in vitro aspect, rather than investigating the phosphorylated status of SIRT3 (which is still a matter of debate in the literature), we investigated the acetylation status of SOD2 and IDH2 in respect to our in vivo results. Unfortunately, H9C2 cells proved to have very low level of acetylated-

SOD2/IDH2 to measure any effect of STA. The results are provided below for the reviewer and we feel that this is not necessary to add this figure to the main paper.

H9C2 cells were submitted to hypoxia (H) or normoxia (N) in the presence or not of STA (S) for 2h. Cell extracts were immunoprecipitated with an anti-Ac-Lys antibody, and blotted for IDH2 or SOD2. Star indicates antibody light chains, arrowheads indicate specific bands. Bottom panel shows an higher exposure of the SOD2 blot. Saturated pixels are shown in red. Results are representative of two independent experiments. The amount of acetylated-SOD2 was estimated ranging from 0.1-0.6% of the total SOD2.

- Please provide data on the normal diet animals in Figures 6-8, as it is important whether the STA treated groups are more similar to ND or not

We now provide additional data on ND-fed age-matched mice. In the new version of our manuscript we report the effects of STA on left ventricular function, cardiac hypertrophy, myocardial fibrosis (Fig.5A-J), oxidative stress and apoptosis (Fig.6A-E), mitochondrial ultrastructural status and expression of mitochondrial-encoded genes in complexes I, II, III, IV and V in cardiac tissue (Figure 7A-D) in HFD-fed mice vs ND mice. We have revised the text accordingly.

- For the mouse experiments, showing body composition for the HFD/STA group would go a long way to understanding the physiological changes caused by STA

We agree with the reviewer that body composition measurement would have been an interesting point. However, in our experiments we did not find any differences in body weight between vehicle- and STA-treated HFD mice, as stated in the results' section. In the same line, we found that STA did not change the amount of perigonadal adipose tissue: (2.27% +/- 0.16 in vehicle-treated mice vs 2.20% +/- 0.11 in STA-treated mice, expressed as % of body weight), suggesting that STA treatment had no major effect on fat depot in obese mice.

Minor points

- Is the upper the upper band in panel 4C nonspecific? If so, could you please provide evidence and mark it as such with an asterisk.

The documented size for processed SIRT3 is around 28kDa (Schwer et al, 2002) and is consistent with our blots (see Figure 3C on isolated mitochondria from H9C2 and Figure 8 and 9D on mouse heart lysates). We provide below additional western-blot highlighting the aspecific band detected by the anti-SIRT3 antibody only in rat cells and not in mouse extracts. We have annotated Figure 3C with a star to indicate this non-specific band and apologize for the omission.

H9C2 total cell lysates or mouse heart extracts were immunoblotted with an anti-SIRT3 antibody. Arrowheads indicate specific ~28kDa band for endogenous SIRT3 while star indicate non-specific band in rat H9C2 cells.

- Two way ANOVA's testing for interactions between STA and stress (or STA/stress/knockdown) may be more appropriate to evaluate these data.

We have performed one-way or two-way ANOVA followed by Bonferroni's post-hoc test when appropriate to analyze the data. Figures' legends and Methods section have been amended accordingly.

Referee #3 (Remarks):

In this manuscript, the authors described a lipid kinase PIKfyve as a regulator of cardiometabolic status and mitochondrial integrity using a diet-induced obesity mouse model. In vitro and in vivo experiments were employed to demonstrate the effect of PIKfyve inhibition on the reverse of obesity -induced cardiac mitochondrial damage and apoptosis potentially through SIRT3 signaling. While the results provided some insights into the potential mechanisms related to obesity-associated myocardial abnormalities, for the most part, the experiments described only associations among some potential pathways. The manuscript has the following major limitations that the authors should consider for future submission:

1) The limitation on the obesity model: while most of the diet-induced obesity model using a 45% high fat diet only need 5 months for the development of obesity, why did the authors choose to feed the mice for 12 months which is a very long period compared to most of the published obesity models? How do the authors distinguish the effects between obesity and aging?

We would like to thank the reviewer for his careful reading and suggestions on our manuscript. Recently, we have shown that prolonged exposure to a HFD for 18 weeks results in functional cardiac adaptation in mice (Alfarano et al, 2014). Indeed, subsequent echocardiographic analysis revealed increased LV posterior wall thickness and interventricular septum thickness without significant changes in fractional shortening and ejection fraction. In agreement with echocardiographic data, the expression levels of two well-known marker genes for impaired LV function, atrial natriuretic peptide and brain natriuretic peptide, were unchanged in HFD-fed mice as compared to ND-fed mice (Alfarano et al, 2014). In order to induce chronic heart failure, in the present study mice were chronically exposed to a HFD for a total of 12 months. Echocardiographic analyses confirmed the obesity-induced cardiac phenotype (Figure 5B-E).

In order to rule out any effect of STA on heart aging/senescence, we have performed qPCR analysis for the expression of the senescence marker p16^{INK4a}. The results, presented below, clearly show that STA did not affect heart senescence in HFD- 12 month mice. We feel that this is not necessary to add this figure to the main paper but if the reviewer thinks otherwise we are willing to do so.

STA treatment does not affect heart aging in HFD-12 months mice. qPCR analysis for the cardiac expression of the senescence marker p16^{INK4a} from ND or HFD mice of the indicated age. ns, non-significant.

2) The authors totally ignored the diastolic dysfunction associated with high fat diet feeding. This is a major flaw of the experimental design, as diabetes is first associated with diastolic abnormalities.

Indeed, the incidence of diastolic dysfunction is high in type 2 diabetes subjects. Diastolic dysfunction has been described as an early sign of the diabetic heart muscle disease preceding the systolic damage (Cosson & Kevorkian, 2003; Zabalgoitia et al, 2001). In the clinical realm, the diabetic patients frequently have evidence of LV diastolic dysfunction with preserved systolic function (Sharma & Kass, 2014). Heart failure can occur in these patients as a result of impaired

ventricular relaxation, requiring elevated filling pressures to obtain normal LV end-diastolic volumes. In the present study, we found that left ventricular systolic function was severely impaired in mice exposed to HFD for 12 months. Thus, our *in vivo* study was focused on the effects of STA on structural and functional properties of the myocardium in a mouse model of heart failure with systolic dysfunction using an echo-guided technique. For an exact determination of diastolic dysfunction LV catheterization would have been required.

3) As PIKfyve is ubiquitously expressed, in which cell type the inhibition with STA affords the protection? In order to determine any effects related to PIKfyve, data on the expression of PIKfyve should be presented.

We present below data showing the expression of PIKfyve in H9C2 cells, isolated cardiomyocytes and mice heart extracts. PIKfyve expression in cardiomyoblasts is further supported by the fact that STA treatment induced the formation of enlarged vacuoles, a hallmark for PIKfyve inhibition (see Figure 1C). As PIKfyve expression has been previously described in cardiac tissue (Ikonov et al, 2013), we do not feel necessary to add this figure in the manuscript and we have amended the introduction section accordingly.

4) The high fat diet-induced obesity is a chronic progression, especially in the current manuscript, how could the acute treatment with STA for 17 days have such a long-lasting chronic effect to reverse all the effects associated with obesity?

In our study the mice were submitted to 12 months of HFD, and only then subjected to daily intraperitoneal injections of vehicle or STA. Our data suggest that STA treatment for 17 days of chronic diet-induced obese mice with established heart failure reduces cardiac hypertrophy and preserves left ventricular function. We have rephrased the methods' section and hope to have clarify the experimental protocol.

5) There are a number of serious issues related to the techniques used including IHC staining, western blotting, echo, ...

a. The IHC staining and related conclusions are not convincing at all. The image resolutions are very poor and any conclusions from those images are doubtful.

We apologize for the poor resolution of the previous images. We now provide high resolution IHC images in Figure 5J.

b. The authors need to provide the 2-D M-mode images as the frequency used is relatively low and a convincing resolution should be presented.

We now provide 2-D M-mode images in Figure 5A.

c. The western blots need appropriate ND controls; and most importantly the proteins should be blotted on the same membrane and displayed so. The authors also should provide a rationale on using IP to detect the phosphor-SIRT3.

We now provide ND controls for western blots in Figures 5 and 7 and western blot in Figure 7C is now shown as complete membrane.

Regarding the phosphorylation status of SIRT3: The regulation of SIRT3 activity by phosphorylation is quite recent (Liu et al, 2015) and there is no commercial antibody available against the phospho-SIRT3. Therefore, we resorted to the use of immunoprecipitation of SIRT3 followed by immunoblotting using a phospho-serine antibody.

The source images for western blots are now provided in the revised version of the manuscript (see files Source Data Fig 2-3-7-8-9) to ensure that samples were loaded on the same membrane.

d. The cytokines concentration should be provided.

In addition to myocardial expression of several pro-inflammatory cytokines (Figure EV4A), in the new version of the manuscript, we have measured plasma level of IL-6 and TNF- α , two major inflammatory cytokines. We did not find statistically significant differences of plasma cytokine levels between vehicle- and STA-treated HFD-fed mice (please, see Figure EV4B). This is consistent with our qPCR data in Figure EV4A and in line with previous results obtained *in vitro* on human blood cells and on skin biopsies from patients orally treated with STA in the context of psoriasis disease (Wada et al, 2012). Results section has been amended accordingly.

References:

- Ahn BH, Kim HS, Song S, Lee IH, Liu J, Vassilopoulos A, Deng CX, Finkel T (2008) A role for the mitochondrial deacetylase Sirt3 in regulating energy homeostasis. *Proc Natl Acad Sci U S A* **105**(38): 14447-14452
- Alfarano C, Foussal C, Lairez O, Calise D, Attane C, Anesia R, Daviaud D, Wanecq E, Parini A, Valet P, Kunduzova O (2014) Transition from metabolic adaptation to maladaptation of the heart in obesity: role of apelin. *Int J Obes (Lond)* **39**(2): 312-320
- Bell EL, Guarente L (2011) The SirT3 divining rod points to oxidative stress. *Mol Cell* **42**(5): 561-568
- Bournat JC, Brown CW (2010) Mitochondrial dysfunction in obesity. *Curr Opin Endocrinol Diabetes Obes* **17**(5): 446-452
- Cai X, Xu Y, Kim YM, Loureiro J, Huang Q (2014) PIKfyve, a class III lipid kinase, is required for TLR-induced type I IFN production via modulation of ATF3. *J Immunol* **192**(7): 3383-3389
- Cosson S, Kevorkian JP (2003) Left ventricular diastolic dysfunction: an early sign of diabetic cardiomyopathy? *Diabetes Metab* **29**(5): 455-466
- Dupuis-Coronas S, Lagarrigue F, Ramel D, Chicanne G, Saland E, Gaits-Iacovoni F, Payrastré B, Tronchère H (2011) The nucleophosmin-anaplastic lymphoma kinase oncogene interacts, activates, and uses the kinase PIKfyve to increase invasiveness. *J Biol Chem* **286**(37): 32105-32114
- Ikonomov OC, Sbrissa D, Delvecchio K, Feng HZ, Cartee GD, Jin JP, Shisheva A (2013) Muscle-specific Pikfyve gene disruption causes glucose intolerance, insulin resistance, adiposity, and hyperinsulinemia but not muscle fiber-type switching. *Am J Physiol Endocrinol Metab* **305**(1): E119-131
- Ikonomov OC, Sbrissa D, Shisheva A (2006) Localized PtdIns 3,5-P2 synthesis to regulate early endosome dynamics and fusion. *Am J Physiol Cell Physiol* **291**(2): C393-404
- Kim HS, Patel K, Muldoon-Jacobs K, Bisht KS, Aykin-Burns N, Pennington JD, van der Meer R, Nguyen P, Savage J, Owens KM, Vassilopoulos A, Ozden O, Park SH, Singh KK, Abdulkadir SA, Spitz DR, Deng CX, Gius D (2010) SIRT3 is a mitochondria-localized tumor suppressor required for maintenance of mitochondrial integrity and metabolism during stress. *Cancer Cell* **17**(1): 41-52
- Kim SM, Roy SG, Chen B, Nguyen TM, McMonigle RJ, McCracken AN, Zhang Y, Kofuji S, Hou J, Selwan E, Finicle BT, Nguyen TT, Ravi A, Ramirez MU, Wiher T, Guenther GG, Kono M, Sasaki AT, Weisman LS, Potma EO, Tromberg BJ, Edwards RA, Hanessian S, Edinger AL (2016) Targeting cancer metabolism by simultaneously disrupting parallel nutrient access pathways. *J Clin Invest* **126**(11): 4088-4102
- Liu R, Fan M, Candas D, Qin L, Zhang X, Eldridge A, Zou JX, Zhang T, Juma S, Jin C, Li RF, Perks J, Sun LQ, Vaughan AT, Hai CX, Gius DR, Li JJ (2015) CDK1-Mediated SIRT3 Activation Enhances Mitochondrial Function and Tumor Radioresistance. *Mol Cancer Ther* **14**(9): 2090-2102
- McDonnell E, Peterson BS, Bomze HM, Hirschey MD (2015) SIRT3 regulates progression and development of diseases of aging. *Trends Endocrinol Metab*
- Morris JB, Hinchliffe KA, Ciruela A, Letcher AJ, Irvine RF (2000) Thrombin stimulation of platelets causes an increase in phosphatidylinositol 5-phosphate revealed by mass assay. *FEBS Lett* **475**(1): 57-60

Niebuhr K, Giuriato S, Pedron T, Philpott DJ, Gaits F, Sable J, Sheetz MP, Parsot C, Sansonetti PJ, Payraastre B (2002) Conversion of PtdIns(4,5)P(2) into PtdIns(5)P by the S.flexneri effector IpgD reorganizes host cell morphology. *EMBO J* **21**(19): 5069-5078

Samant SA, Zhang HJ, Hong Z, Pillai VB, Sundaresan NR, Wolfgeher D, Archer SL, Chan DC, Gupta MP (2014) SIRT3 deacetylates and activates OPA1 to regulate mitochondrial dynamics during stress. *Mol Cell Biol* **34**(5): 807-819

Sarkes D and Rameh LE (2010) A novel HPLC-based approach makes possible the spatial characterization of cellular PtdIns5P and other phosphoinositides. *Biochem J* **428**(3): 375-384

Schwer B, North BJ, Frye RA, Ott M, Verdin E (2002) The human silent information regulator (Sir)2 homologue hSIRT3 is a mitochondrial nicotinamide adenine dinucleotide-dependent deacetylase. *J Cell Biol* **158**(4): 647-657

Sharma K, Kass DA (2014) Heart failure with preserved ejection fraction: mechanisms, clinical features, and therapies. *Circ Res* **115**(1): 79-96

Tsutsui H, Kinugawa S, Matsushima S (2011) Oxidative stress and heart failure. *Am J Physiol Heart Circ Physiol* **301**(6): H2181-2190

Wada Y, Cardinale I, Khatcherian A, Chu J, Kantor AB, Gottlieb AB, Tatsuta N, Jacobson E, Barsoum J, Krueger JG (2012) Apilimod inhibits the production of IL-12 and IL-23 and reduces dendritic cell infiltration in psoriasis. *PLoS One* **7**(4): e35069

Weir HJ, Lane JD, Balthasar N (2013) SIRT3: A Central Regulator of Mitochondrial Adaptation in Health and Disease. *Genes Cancer* **4**(3-4): 118-124

Zabaloitia M, Ismaeil MF, Anderson L, Maklady FA (2001) Prevalence of diastolic dysfunction in normotensive, asymptomatic patients with well-controlled type 2 diabetes mellitus. *Am J Cardiol* **87**(3): 320-323

Zolov SN, Bridges D, Zhang Y, Lee WW, Riehle E, Verma R, Lenk GM, Converso-Baran K, Weide T, Albin RL, Saltiel AR, Meisler MH, Russell MW, Weisman LS (2012) In vivo, PIKfyve generates PI(3,5)P2, which serves as both a signaling lipid and the major precursor for PI5P. *Proc Natl Acad Sci U S A* **109**(43): 17472-17477

2nd Editorial Decision

07 March 2017

Thank you for the submission of your revised manuscript to EMBO Molecular Medicine. We have now received the enclosed reports from the referees that were asked to re-assess it. As you will see the reviewers are now supportive and I am pleased to inform you that we will be able to accept your manuscript pending the following final amendments:

1) Please address the minor text change commented by referee 2. Please provide a letter INCLUDING the reviewer's reports and your detailed responses to their comments (as Word file).

Please submit your revised manuscript within two weeks. I look forward to seeing a revised form of your manuscript as soon as possible.

***** Reviewer's comments *****

Referee #1 (Remarks):

In this resubmitted manuscript, Tronchere et al show that PIKfyve controls mitochondrial fragmentation, hypertrophic and apoptotic responses to stress, in part by activating SIRT3. By completing additional experiments, this study is further strengthened and I have no other comments to offer.

Referee #2 (Comments on Novelty/Model System):

There appears to be some lack of congruence between the cell culture and animal model systems, but the authors seem to have addressed many of these concerns in the revised version.

Referee #2 (Remarks):

The new data presented in this version, especially the SIRT3 knockout and PIKfyve knockout

studies are compelling to support the previous claims of a cardioprotective effect of PIKfyve inhibition on HFD and hypoxia-induced cardiac dysfunction. The authors should be commended for making such rapid alterations and improvements in this manuscript. I offer only minor suggestions for clarification.

In terms of the basal PI5P being unaffected by STA treatment, this does not necessarily indicate responsibility for basal turnover, but rather that basal levels of PI5P are refractory to inhibition by STA treatment.

The lack of detectable change in fat pad mass in the HFD/STA treated animals vs the HFD animals seems important enough to include in the manuscript, as it indicates this is not a general resolution of adiposity.

Referee #3 (Remarks):

This reviewer finds the revision has addressed all the concerns and it is suitable for publication now.

2nd Revision - authors' response

14 March 2017

Referee #1 (Remarks):

In this resubmitted manuscript, Tronchere et al show that PIKfyve controls mitochondrial fragmentation, hypertrophic and apoptotic responses to stress, in part by activating SIRT3. By completing additional experiments, this study is further strengthened and I have no other comments to offer.

Referee #2 (Comments on Novelty/Model System):

There appears to be some lack of congruence between the cell culture and animal model systems, but the authors seem to have addressed many of these concerns in the revised version.

Referee #2 (Remarks):

The new data presented in this version, especially the SIRT3 knockout and PIKfyve knockout studies are compelling to support the previous claims of a cardioprotective effect of PIKfyve inhibition on HFD and hypoxia-induced cardiac dysfunction. The authors should be commended for making such rapid alterations and improvements in this manuscript. I offer only minor suggestions for clarification.

In terms of the basal PI5P being unaffected by STA treatment, this does not necessarily indicate responsibility for basal turnover, but rather that basal levels of PI5P are refractory to inhibition by STA treatment.

We thank the reviewer for his analysis and suggestion. We have amended the Results' section accordingly.

The lack of detectable change in fat pad mass in the HFD/STA treated animals vs the HFD animals seems important enough to include in the manuscript, as it indicates this is not a general resolution of adiposity.

We totally agree with the reviewer that this is an interesting point. We have included these data in the manuscript.

Referee #3 (Remarks):

This reviewer finds the revision has addressed all the concerns and it is suitable for publication now.

Corresponding Author Name: Frederic Boal
Journal Submitted to: EMBO Mol Med
Manuscript Number: EMM-2016-07096-V2 (Inhibition of PIKfyve prevents myocardial apo...)